# Winds and Tides of the Extended Unified Model in the Mesosphere and Lower Thermosphere Validated with Meteor Radar Observations

Matthew J. Griffith[1], Shaun M. Dempsey[2], David R. Jackson[3], Tracy Moffat-Griffin[4], and Nicholas J. Mitchell[2,4]

[1]Department of Mathematical Sciences, University of Bath, Claverton Down, Bath, BA2 7AY, UK.
[2]Department of Electronic & Electrical Engineering, University of Bath, Claverton Down, Bath, BA2 7AY, United Kingdom.
[3]Met Office, Fitzroy Rd, Exeter, EX1 3PB, United Kingdom.
[4]British Antarctic Survey, High Cross, Madingley Rd, Cambridge, CB3 0ET, United Kingdom.

**Correspondence:** Matthew Griffith (M.J.Griffith@bath.ac.uk)

**Abstract.** The Mesosphere and Lower Thermosphere (MLT) is a critical region that must be accurately reproduced in General Circulation Models (GCMs) that aim to include the coupling between the lower & middle atmosphere and the thermosphere. An accurate representation of the MLT is thus important for improved climate modelling and the development of a whole atmosphere model. This is because the atmospheric waves at these heights are particularly large, and so the energy and momentum they carry is an important driver of climatological phenomena through the whole atmosphere, affecting terrestrial and space weather. The Extended Unified Model (ExUM) is the recently developed version of the Met Office's Unified Model which has been extended to model the MLT. The capability of the ExUM to model atmospheric winds and tides in the MLT is currently unknown. Here, we present the first study of winds & tides from the ExUM. We make a comparison against meteor radar observations of winds and tides from 2006 between 80 and 100 km over two radar stations – Rothera (68° S, 68° W) and Ascension Island (8° S, 14° W). These locations are chosen to study tides in two very different tidal regimes – the equatorial regime, where the diurnal (24 hour) tide dominates, and the polar regime, where the semi-diurnal (12 hour) tide dominates. The results of this study illustrate that the ExUM is capable of reproducing atmospheric winds and tides that capture many of the key characteristics seen in meteor radar observations, such as zonal & meridional wind maxima and minima, the increase in tidal amplitude with increasing height, and the decrease in tidal phase with increasing height. In particular, in the equatorial regime some essential characteristics of the background winds, tidal amplitudes and tidal phases are well captured, but with significant differences in detail. In the polar regime, the difference is more pronounced. The ExUM zonal background winds in austral winter are primarily westward rather than eastward, and in austral summer are larger than observed above 90 km. The ExUM tidal amplitudes here are in general consistent with observed values, but are also larger than observed values above 90 km in austral summer. The tidal phases are generally well replicated in this regime. We propose that the bias in background winds in the polar regime is a consequence of the lack of in-situ gravity wave generation to generate eastward fluxes in the MLT. The results of this study indicate that the ExUM has a good natural capability for modelling atmospheric winds and tides in the MLT, but that there is room for improvement in the model physics in this region. This highlights the need for modifica-

tions to the physical parameterization schemes used in the model in this region – such as the non-orographic spectral gravity wave scheme – to improve aspects such as polar circulation. To this end, we make specific recommendations of changes that can be implemented to improve the accuracy of the ExUM in the MLT.

## 1 Introduction

Atmospheric solar tides are global-scale oscillations of the atmosphere. They are primarily forced by solar heating of water vapour and ozone in the troposphere and stratosphere, by the release of latent heat in deep tropospheric convection or by planetary-scale nonlinear interactions. The tides can ascend to the mesosphere and lower thermosphere (MLT) where they reach very large amplitudes and often dominate the motion field. Observations have revealed that the largest-amplitude tidal modes in the MLT are the 12 hour semi-diurnal and 24 hour diurnal tides. Tides of higher-frequency are usually of much smaller amplitudes. Generally, the semi-diurnal tide maximises at high latitudes, whereas the diurnal tide maximises at low latitudes (Mitchell et al., 2002; Davis et al., 2013).

The tides can have significant fluxes of energy and momentum and so play a critical role in coupling the lower atmosphere and the thermosphere-ionosphere system. For example, tidal winds act to filter the field of atmospheric gravity waves (GWs), modulating the gravity-wave momentum fluxes and the consequent forcing of the global atmospheric circulation (e.g., Fritts and Alexander, 2003). The temperature tides are believed to be an important source of the variability of polar mesospheric clouds, because tidal perturbations of temperature modulate the cloud ice crystal population (e.g., Fiedler et al., 2005). Tidal signatures propagate upwards from the MLT into the thermosphere, where the divergence of tidal momentum and heat fluxes can drive zonal wind changes of more than $30 \, \mathrm{ms^{-1}}$ in the lower thermosphere and influence the transport of chemical species (e.g., Jones Jr et al., 2014). Tides are also generated in-situ in the thermosphere from the dissipation of GWs cause by deep convection, primarily in the Intertropical Convergence Zone (ITCZ) (Vadas et al., 2014). These thermospheric tides cause perturbations of neutral and plasma densities in the E- and F-regions of the ionosphere and so modulate the ionospheric wind dynamo (e.g., Oberheide et al., 2009; Yiğit and Medvedev, 2015; Liu, 2016). They can also change the Total Electron Content (TEC) distributions, with the consequence of potentially modifying the conditions for seeding of equatorial plasma bubbles in the F-region. The significant zonal wavenumber-four structure in the equatorial ionosphere is thought to arise from the modulation of the E-region winds by a non-migrating diurnal tide (e.g., England et al., 2006).

The tides include both migrating and non-migrating modes. The migrating modes are sun-synchronous, propagate westwards, have zonal wavenumbers equal to the number of cycles of the tide per day and are directly excited by the insolation of solar radiation. In contrast, the non-migrating modes are not sun-synchronous, can propagate both eastwards or westwards and have zonal wavenumbers not equal to the number of cycles of the tide per day. The non-migrating modes can be excited by strong non-linear interaction between migrating tides and planetary waves that generate so-called "secondary waves", including the non-migrating tidal modes (e.g., Teitelbaum and Vial, 1991; Beard et al., 1999; Palo et al., 2007; Pancheva et al., 2002). The tides and waves of the MLT consequently form a strongly coupled system and at any point in the atmosphere the tides are a superposition of both migrating and non-migrating modes.

A striking feature of atmospheric tides is their variability on a wide range of time scales. For instance, tidal amplitudes and phases have been observed to have a strong seasonal variability (e.g., Mitchell et al., 2002; Davis et al., 2013). This has been proposed to result from phenomena including wave/mean-flow interactions and/or source variations and refraction/reflection (e.g., McLandress, 2002; Riggin et al., 2003; Riggin and Lieberman, 2013). Intra-seasonal variability is also observed. For example, variability of Arctic semi-diurnal tides has been shown to be well correlated with the amplitude of planetary wavenumber 1 at Antarctic latitudes, indicating significant inter-hemispheric coupling (Smith et al., 2007). At inter-annual time scales, tidal amplitudes and phases have been observed to vary in response to solar variability, the El Niño Southern Oscillation, sudden stratospheric warmings, the tropical Madden-Julian Oscillation and the stratospheric Quasi-Biennial Oscillation (e.g., Christiansen et al., 2016).

At time scales of less than 30 days, the tides are observed to exhibit great variability and amplitudes are frequently observed to fluctuate from day to day by up to about 300 % (e.g., Dempsey et al., 2021; Vitharana et al., 2019). This "tidal weather" has been proposed to have causes that include, i) variations in the background winds through which the tides must propagate, ii) variations in tidal forcing resulting from solar variability and/or fluctuations in the distribution of water vapour and stratospheric ozone (e.g., Pancheva and Mitchell, 2004; Lieberman et al., 2004) and iii) non-linear interactions with planetary waves that generate secondary waves that then beat with the primary tide, modulating its amplitude (e.g., Teitelbaum and Vial, 1991; Beard et al., 1999; Palo et al., 2007).

There is now considerable scientific interest in developing so-called "high-top" General Circulation Models (GCMs) that span the lower, middle and upper atmosphere, capturing vertical coupling processes via internal waves such as gravity waves and tides (e.g., Yiğit et al., 2016). These models are an important element in attempts to develop operational space-weather forecasting able to include the contributions to the variability of the thermosphere and ionosphere, as well as in the development of Whole Atmosphere Models (e.g., Jackson et al., 2019; Liu, 2016; Akmaev, 2011). We summarise some of the recent key non-mechanistic "high-top" GCMs below:

1. The Whole Atmosphere Model (WAM; Akmaev et al., 2008; Fuller-Rowell et al., 2008) is an extended version of the U.S National Weather Service Numerical Weather Prediction model, spanning the surface to around 600 km. Focusing on the neutral atmosphere, WAM is able to represent well the mean state and tides in the thermosphere (e.g. Lieberman et al. (2013) show good agreement with diurnal and time mean Challenging Mini Satellite Payload winds).

2. The Whole Atmosphere Community Climate Model with thermosphere and ionosphere extension (WACCM-X; Liu et al., 2010, 2018a) is an extended version of the National Center for Atmospheric Research's WACCM, which itself can run up to 145 km (e.g., Garcia et al., 2007). It has a similar altitude range to WAM. Liu et al. (2018a) show that in WACCM-X the amplitudes and seasonal variations of atmospheric tides in the MLT are in good agreement with observations.

3. The extended Canadian Middle Atmosphere Model (eCMAM; Beagley et al., 2000) is an extended version of the standard CMAM with an upper boundary at a pressure level of $2 \times 10^{-7}$ hPa. eCMAM was developed to examine the nature of the physics and dynamical processes in the MLT without the artificial effects of a sponge layer, which can

have the unfortunate effect of modifying the circulation in the model in an unrealistic fashion (Fomichev et al., 2002). Dempsey et al. (2021) show that eCMAM generally reproduces observed diurnal tidal amplitudes in the polar regime well, and Davis et al. (2013) show that eCMAM is generally good in the equatorial regime with a trend of overestimating meridional amplitudes.

4. The Ground-to-topside model of the Atmosphere and Ionosphere for Aeronomy (GAIA; Fujiwara and Miyoshi, 2010; Jin et al., 2012, and references therein) combines three independent models: a whole atmosphere GCM, an ionosphere model, and an electrodynamics model. GAIA also has a similar altitude range to WAM. Jin et al. (2012) show the ability of GAIA to model the impact of an SSW on migrating tides and the associated ionospheric response, with in general good agreement shown with Sounding of the Atmosphere using Broadband Emission and Constellation Observing System for Meteorology, Ionosphere, and Climate observations.

5. The Hamburg Model of the Neutral and Ionized Atmosphere (HAMMONIA; Schmidt et al., 2006; Meraner and Schmidt, 2016) is an extended version of MAECHAM5 (Giorgetta et al., 2006; Manzini et al., 2006), taking the upper boundary to approximately 250 km. The extended model includes important radiative and dynamical processes of the upper atmosphere and is coupled to a chemistry module containing 48 compounds.

6. The upper-atmosphere extension of ICON (Borchert et al., 2019) extends the standard ICON model so that model upper boundaries can be placed in the lower thermosphere. This includes a switch over to deep-atmosphere dynamics, as well as an implementation of an upper-atmosphere physics package based on that implemented by Schmidt et al. (2006) in HAMMONIA.

7. The Entire Atmosphere GLobal model (EAGLE; Klimenko et al., 2019) combines the HAMMONIA neutral atmosphere model with the Global Self-consistent Model of the Thermosphere, Ionosphere, Protonosphere (GSM TIP) (Bessarab et al., 2012; Korenkov et al., 2012). The model includes radiative heating due to absorption of extreme solar UV, non-LTE treatment of the radiative cooling, molecular diffusion, ion drag, as well as simplified ion chemistry with which to treat the impact of precipitating energetic particles. Klimenko et al. (2019) show that the model successfully reproduces neutral temperature and total electron content (TEC) observations.

8. The HI Altitude Mechanistic General Circulation Model (HIAMCM; Becker and Vadas, 2020) is an extension of the high-resolution Kühlungsborn Mechanistic general Circulation Model (KMCM) extended to around 450 km. The model includes simplified but nevertheless explicit representations of the relevant components of an atmospheric climate model, and is labelled "mechanistic" due its use of some idealized methods and the lack of a chemistry scheme. It is a high resolution gravity wave resolving model, resolving horizontal wavelengths down to 165 km. Becker and Vadas (2020) showed that this GCM is unique in reproducing the travelling atmospheric disturbance (TAD) hotspot observed over the wintertime Southern Andes (e.g., Park et al., 2014; Trinh et al., 2018).

9. The Coupled Middle Atmosphere Thermosphere-2 (CMAT-2; Yiğit et al., 2009) GCM is an extension of the three-dimensional Coupled Thermosphere-Ionosphere-Plasmasphere model (CTIP; Fuller-Rowell et al., 1996; Millward et al.,

1996) to an upper boundary of 300 - 500 km, depending on the solar activity. It uses a nonlinear spectral GW parameterization of Yiğit et al. (2008) to study the propagation of a broad spectrum of GWs from the lower atmosphere to the thermosphere. Yiğit et al. (2021) found that accounting for latitudinal variations in the GW source appreciably improves simulations.

10. The University of Leipzig Middle and Upper Atmosphere Model (MUAM; Pogoreltsev, 2007; Pogoreltsev et al., 2007; Suvorova and Pogoreltsev, 2011) extends from the lower atmosphere up to 160 km. In a recent study by Lilienthal et al. (2020) with this GCM on the interaction of GW and terdiurnal tides they found a strong dependence of tidal amplitude on the induced GW drag, generally being larger when GW drag is increased, whilst the overall strength of the GW source level momentum flux had a relatively small impact on the zonal mean climatology.

11. The whole atmosphere Kyushu GCM (Miyoshi and Fujiwara, 2008; Miyoshi and Yiğit, 2019) extends the pre-exisiting Kyushu GCM (Miyahara et al., 1993) up to 450 km. Miyoshi and Yiğit (2019) found that GW drag in the thermosphere significantly decelerates the mean zonal wind and plays an important role in the momentum budget, making a GW parameterization accounting for thermospheric processes essential for a coarse-grid whole atmosphere GCM.

In the context of these existing models, the Extended Unified Model (ExUM; Griffith et al., 2020) described in Sect. 2.1 extends the standard UM (Unified Model) (Walters et al., 2017) to the lower thermosphere. It is a model which does not make the hydrostatic assumption and uses the deep-atmosphere equations of motion making it particularly suitable for modelling atmospheric tides. As well as this, a non-LTE radiation scheme has been added so that the radiation scheme is physically appropriate up to 90 km (see Jackson et al., 2020) and after this the temperature is nudged toward an analytical profile – see Sect. 2.1 or Griffith et al. (2020) for more details.

Throughout these models and studies, tides have an important role in coupling the lower and upper atmosphere and it is important that tides are accurately represented. However, it is widely recognised that tides in the MLT remain challenging to model (e.g., Baldwin et al., 2019). In particular, model biases remain in both the seasonal variability of tides and their short-term variability at time scales of less than a month (e.g., Dempsey et al., 2021).

As well as tides, it is important that the deposition of momentum by sub-grid scale non-orographic GWs is also accurately represented in models through parameterization, due to their appreciable impact on atmospheric flow and tides in the MLT (e.g., Yiğit and Medvedev, 2017; Yiğit et al., 2009; Miyahara and Forbes, 1991). Yiğit and Medvedev (2017) discuss extensively the influence of parameterized small-scale GWs on the migrating diurnal tide. They show that GWs play an important role for the diurnal tide in the MLT region. They found that the GW effects on the thermal tide can be appropriately captured in a coarse-grid GCM provided that a GW parameterization, i) considers a broad spectrum of harmonics, ii) properly describes their propagation, and iii) correctly accounts for the physics of wave breaking/saturation. Yiğit et al. (2021) suggest that smaller than measured GW fluxes have to be used at the source level in the lower atmosphere in order to reproduce the observed circulation in the middle atmosphere.

In this study, we test the ability of the ExUM to model diurnal and semi-diurnal tides by comparing the seasonal variation of these tides in the model to observations of zonal and meridional winds made in the mesosphere and lower thermosphere by

two meteor radars. The two radars are at very different latitudes, one at the polar Antarctic site of Rothera (68° S, 68° W) and the other at Ascension Island (8° S, 14° W) in the equatorial Atlantic Ocean. The Rothera radar samples a latitude where the semi-diurnal tide is known to reach very large amplitudes but where the diurnal tide is small. In contrast, the Ascension Island radar samples a region where the diurnal tide is known to reach large amplitudes but the semi-diurnal tide is small. We use measurements of winds, tidal amplitudes, tidal phases and their seasonal variability as tests of the model's ability to accurately represent these tides.

The meteor radars are particularly well suited for this task because they can make continuous reliable measurements at the heights of 80–100 km where the tidal modes reach large amplitude, but where other ground-based radar measurement techniques, such as MF radar, may be subject to significant biases (e.g., Wilhelm et al., 2017). In fact, a recent study by Stober et al. (2021) examined the mean winds, diurnal & semi-diurnal tidal amplitudes & phases and their associated momentum fluxes obtained from meteor radar data at six Southern Hemisphere locations (midlatitude to polar). They found that the results agreed reasonably well with Becker and Vadas (2018), thereby pointing to secondary GWs and vertical coupling as a mechanism by which GWs transfer energy and momentum to higher altitudes under wintertime conditions.

This study is organised as follows. In Sect. 2 we describe the development of the ExUM version used in this study and the meteor radar observations used to provide the observational "ground truth". In Sect. 3 we present the seasonal variability of background winds and diurnal and semi-diurnal tides in the ExUM and in observations, highlighting areas of agreement and disagreement. Finally, in Sect. 4 and Sect. 5 we place our results in the context of other tidal studies and consider how they can guide future development of the ExUM.

## 2   Model development and meteor radar observations

### 2.1   The Extended Unified Model

The Met Office's Unified Model (UM) is a GCM modelling the weather and climate of the atmosphere. It is split into two main sections; the first contains the *dynamical core*, which describes atmospheric dynamics by numerically solving the Euler equations of motion governing atmospheric flow; and the second is made up of *physical parameterizations*, which attempt to describe parts of atmospheric physics not captured by the governing equations, such as solar radiation and sub-grid scale GWs (see e.g. Walters et al. (2017) for an idea on the formulation of the Unified Model).

The current dynamical core (ENDGame; Wood et al., 2014) solves the non-hydrostatic, fully compressible deep-atmosphere equations of motion on a rotating sphere using a semi-implicit semi-Lagrangian formulation. The primary prognostic variables used are the three-dimensional wind components, virtual dry potential temperature[1], Exner function of pressure[2] and dry

---

[1]The potential temperature $\theta$ is the temperature that an unsaturated parcel of dry air would have if brought adiabatically and reversibly from its initial state to a standard pressure, $p_0$, typically 1000 hPa. The virtual dry potential temperature is then the theoretical potential temperature of dry air that would have the same density as moist air.

[2]The Exner function $\Pi$ can be viewed as non-dimensionalized pressure and has the useful relationship that the absolute temperature $T = \theta\Pi$.

density, whilst moisture prognostics are advected as free tracers. The discretised equations are solved using an iterative implicit method – more details of which can also be found in Wood et al. (2014).

For the purposes of this case study, the horizontal resolution is fixed at $1.25°$ N$\times 1.875°$ E – or the so called N96 resolution[3]. The vertical resolution is extended from the 85-level, 85 km configuration of the standard UM using the model implementation of Griffith et al. (2020). This gives the aforementioned ExUM which builds on the standard model to extend the working height of the UM into the lower thermosphere. The work makes it possible for the Unified Model to run in a stable manner with an upper boundary at 100, 120 and 135 km, with promising initial results.

Griffith et al. (2020) investigated the cause of an instability in the previously unstable ExUM. Through a thorough and systematic diagnostic evaluation of both the dynamical core and physical parameterizations used in the model, the root cause of the instability was identified – the radiation scheme. The assumption of Local Thermodynamic Equilibrium (LTE)[4] used in the model is no longer valid on extension of the upper boundary of the model. To enable further testing without the need to completely re-engineer the existing radiation scheme – a significant undertaking – an interim solution of relaxation or nudging of the temperature field to climatological values was used. This scheme was engineered in Griffith et al. (2020) and more details can be found therein. With this addition, a stable ExUM implementation was successfully achieved with upper boundary heights of 100, 120 and 135 km. The 120 and 135 km implementations did however require additional stability modifications such as an increase in the value of the vertical damping coefficient. The primary impact of the damping coefficient is to reduce the magnitude of instantaneous vertical velocities approaching the upper boundary which can lead to model instabilities. The damping coefficient is therefore chosen as the minimal value so that the model can run in a stable manner (for more details on the specifics of the vertical damping used and the implementations for the 120 and 135 km upper boundary please see Griffith et al. (2020) - in particular, Sect. 2.2, Figure 1 and Sect 5.2 therein). Also, the nudging temperature profile used is globally uniform, and so the latitudinal variation in temperature was difficult to attain – for example the summertime polar mesopause minimum was present but not realistically captured. However, in this initial research the focus was to produce a stable extension of the UM, rather than to focus on precisely capturing realistic climatology. Thus, for this stability analysis, a simple and approximate climatological nudging temperature profile was used, which successfully showed that the model could run in a stable manner into the Mesosphere and Lower Thermosphere (MLT).

Following this research, the radiation scheme was extended to include non-LTE effects and the model temperature now contains the appropriate realistic forcing up to around 90 km. This work is detailed by Jackson et al. (2020). The improvement to the summertime polar mesopause minimum and consequent improvement in the wind fields can be seen in Fig. 1.

Above around 90 km, the lack of appropriate high atmosphere chemistry and consequent heating via exothermic reactions means that the temperature profile cannot be assumed to be accurate. Given this lack of appropriate chemistry, the relaxation or nudging scheme must still be used above 90 km and is in place for our simulation. This pushes the model temperature towards a globally uniform temperature field, which can be seen for this region in Fig. 2.

---

[3]The integer N represents the maximum number of zonal 2 grid-point waves that can be represented – thus N96 can represent 96 such waves.

[4]The condition under which matter emits radiation based on its intrinsic properties and its temperature, uninfluenced by the magnitude of any incident radiation.

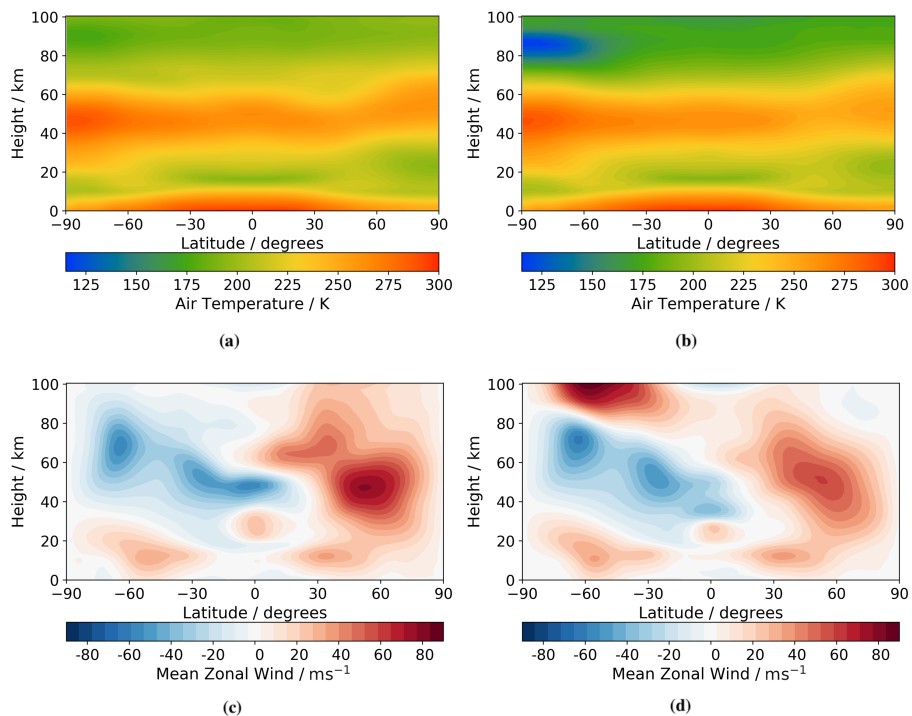

**Figure 1.** Latitude-height zonal-mean monthly-mean climatologies in December comparing **(a)** ExUM temperature before with **(b)** ExUM temperature after the non-LTE implementation. Also compared is **(c)** ExUM zonal ($u$) wind before with **(d)** ExUM zonal ($u$) wind after the non-LTE implementation. The more accurate modelling of the summertime polar mesopause minimum is evident upon introduction of the non-LTE radiation scheme with consequent effect on the modelled winds in the MLT.

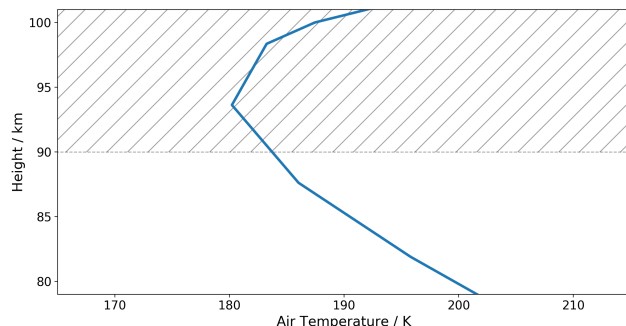

**Figure 2.** Nudging temperature profile over the region of interest sampled on model levels (80–100 km). Shading indicates the height above which this nudging profile is used in the ExUM model run.

In summary, this results in an ExUM which differs from the standard General Atmosphere (GA) 7.0 configuration of the UM (as described in Walters et al., 2017) in the following ways:

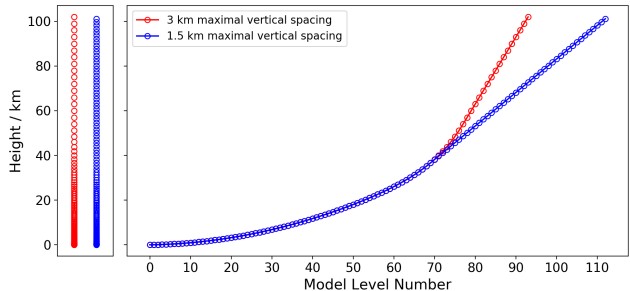

**Figure 3.** Vertical level sets for the 3 km maximal vertical spacing (red) and the 1.5 km maximal vertical spacing (blue).

1. The model chemistry scheme is entirely switched off – the development of a chemistry scheme appropriate for the MLT is currently a work in progress.

2. Atmospheric aerosols are switched off and ozone background files are switched on.

3. The model upper boundary is raised from the standard 85 km to a height of 100 km.

4. The forcing from the radiation scheme now includes non-LTE effects which means it is physically realistic up to 90 km.

5. The temperature field above 90 km is nudged towards the prescribed climatological temperature profile – this accounts for the lack of the chemistry scheme.

With this, the model is now sufficiently mature to ask the question: are the wind fields produced by the new ExUM physically realistic in the mesosphere and lower thermosphere? In this research, we answer this question by performing an initial case study comparing ExUM wind fields and tides to corresponding fields from meteor radar observations.

To begin this case study, we use the work of Griffith et al. (2020) to educate the choice of vertical resolution. The possibilities for a 100 km model upper boundary are a 94-level and 113-level configuration. These are based on the assumption of having a maximum vertical level depth of 3 km and 1.5 km respectively. These depths are based on the atmospheric scale height $H = RT/g$ and give two and four vertical model levels per scale height respectively. These level sets can be seen in Fig. 3.

The 94-level configuration requires no changes to the model's vertical damping coefficient (see Griffith et al. (2020) for more details on the vertical damping coefficient), whereas the 113-level configuration requires a sixfold increase in the vertical damping coefficient used – which can have the undesirable effect of modifying the general circulation in an unrealistic manner (e.g., Fomichev et al., 2002). As well as this, over the two chosen radar locations, the vertical wavelength is typically around 20 km in the MLT (e.g. Davis et al. (2013) Table 1 for Ascension Island and Dempsey et al. (2021) Table 1 for Rothera). The resolution of the radar observations is also 3 km in the vertical, and this is the resolution used in the MLT in other models such as WACCM and eCMAM (described previously).

Therefore, the 94-level configuration is chosen for this study, which has a 3 km vertical resolution in the MLT. It avoids the use of the larger value for the vertical damping coefficient, matches up well with the resolution of the meteor radar observations as well as appropriately resolving wave scales for both radar locations.

Given the above choice of resolution, a choice of start date is required. This is guided by the availability of radar observations and this is discussed in Sect. 2.2 – with 2006 being the year chosen.

The model runs are then all initialised using the same operational analysis from 1 September 2005 at 00 UTC. This allows the model to settle after the initialisation – known as the spin-up period of the model. Following this, climatological data (rather than year dependent data) is used to force background fields such as atmospheric ozone. This choice was made primarily due to the unavailability of year dependent forcing for the recently developed ExUM (such as that used in more developed models like WACCM-X (e.g., Liu et al., 2018b)). The primary focus of this work is to provide a first-look at the atmospheric tides

present in the model, and perform a first comparison of those tides with observations in order to justify that the core dynamics and physics of the model is sound. Differences seen here can then be used to educate future development.

The output attained from the model consists of hourly-sampled time profiles for both zonal and meridional wind fields for the whole of the model year considered – this high cadence is used so that diurnal and semi-diurnal frequencies can be accurately resolved. For simplicity, we only show results for a single simulation, but multiple simulations were performed to verify these

255 results leading to the same conclusions. From these model fields, we compute monthly mean background wind fields and composite days for each month. Each composite day gives an average for each hour of the day over the course of the month at each height in the 80–100 km range being considered. The atmospheric tidal amplitudes and phases are then calculated for each month by fitting a sinusoidal function to this composite day using a curve fitting algorithm.

## 2.2 The meteor radars

We will compare the ExUM model's winds and tides to those measured by meteor radars. Meteor radars are well suited for wind and tidal studies because they can make continuous, reliable, measurements of zonal and meridional winds at the heights of 80–100 km where tidal amplitudes reach large values (e.g., Dempsey et al., 2021). In this particular case, we consider observations made by two commercially-produced all-sky "SKiYMET" radars. One such radar is sited at Rothera (68° S, 68° W) in the Antarctic, a latitude where we expect the semi-diurnal tide to dominate. The other is sited on Ascension Island

(8° S, 14° W) in the equatorial Atlantic, a latitude where we expect the diurnal tide to dominate. The two radars both use the commercially-produced all-sky "SKiYMET" system making their measurements directly comparable. A description of the SKiYMET radar can be found in Hocking et al. (2001). The availability of radar observations for both sites is shown in Fig. 4.

From this, it can be seen that the radars were simultaneously operational with the fewest interruptions throughout 2006, and so we use data from that year in our analysis.

The time series of winds recorded by the radars were analysed to determine tidal amplitudes and phases for the diurnal and semi-diurnal tides. The method employed is essentially a standard least-squares fitting method common in tidal analysis. The particular implementation used here is that described by Dempsey et al. (2021). In this, for each month a composite day of zonal and meridional hourly winds was constructed. A least-squares fit of sinusoidal oscillations with periods of 24, 12, 8 and 6 hour, corresponding to the tides, was then made for each month and each component at each height. The result of this analysis

is a monthly vector mean estimate of the amplitude and phase of each tide at heights from 79–101 km in both the zonal and

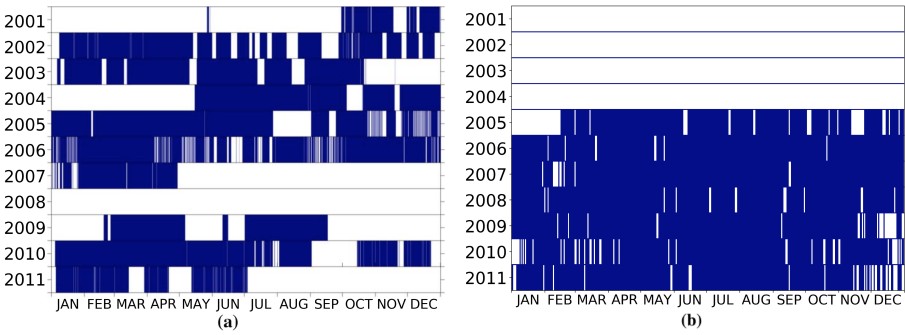

**Figure 4.** Availability of meteor radar observations for **(a)** Ascension Island (from Davis et al., 2013) and **(b)** Rothera. This motivates the choice of 2006 as the year for comparison.

meridional components (we will not consider the 8 and 6 hour tides further in this study). These observed tides can then be compared to those predicted by the ExUM for the two sites.

## 3 Results

In this section, we present the ExUM winds and tides for the latitudes of Rothera and Ascension Island and compare them to the observations made by the two radars. We begin by presenting, in Fig. 5, examples of the ExUM zonal and meridional hourly winds for April 2006. These wind fields are typical of those produced in the MLT by the model and are shown to illustrate the general features of the model results. The figure presents winds for the locations of both Ascension Island and Rothera.

The figure reveals wind fields dominated by tidal modes of large amplitude. As expected, at the Antarctic location of Rothera the semi-diurnal tide dominates, whereas at the equatorial location of Ascension Island the diurnal tide dominates. The ExUM tidal amplitudes display some short-term variability and on occasion reach values in excess of $150 \ \mathrm{ms}^{-1}$. At the location of Ascension Island the ExUM diurnal tidal amplitudes actually decrease slightly at heights above about 90 km. For both locations and in both the zonal and meridional components there is a clear descent of the phase fronts with increasing time, corresponding to upwardly-propagating tides. These tidal oscillations are superposed on background wind fields that themselves display variation in height and time. Before we consider the variability of the tides in more detail, we will thus consider the ExUM's zonal and meridional background winds at the two locations, examine how they vary throughout the year and compare them to the radar observations. All months will be referred to by their 3 letter abbreviation in lists for brevity.

### 3.1 Mean winds

The monthly-mean zonal and meridional winds for Ascension Island are presented in Fig. 6 and those for Rothera in Fig. 7. In each case we also present the corresponding monthly-mean winds observed by the respective radar. The zero-wind line in the figures is indicated by a dashed black line.

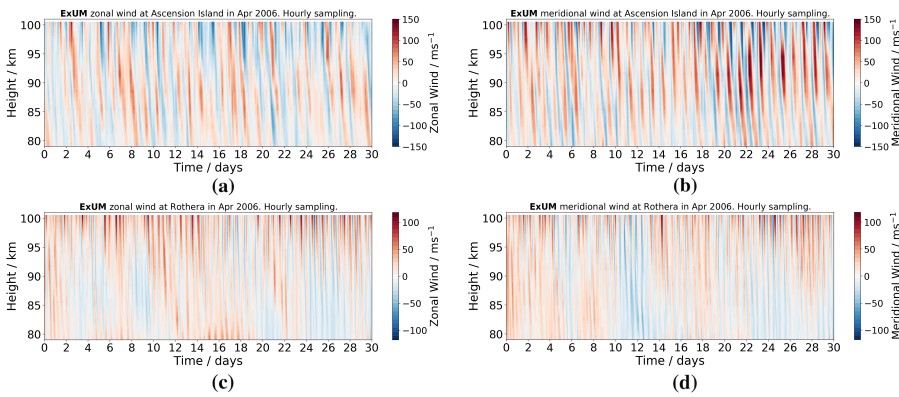

**Figure 5.** ExUM time-height wind contours from the 94-level model configuration in April 2006. At Ascension Island for the **(a)** zonal ($u$) and **(b)** meridional ($v$) components. Similarly at Rothera for the **(c)** zonal ($u$) and **(d)** meridional ($v$) components.

Firstly, we consider the equatorial site of Ascension Island. The ExUM monthly-mean zonal winds clearly exhibit the well-known mesospheric semi-annual oscillation, with wind maxima in January and June and minima in April and October. The amplitude of this semi-annual behaviour reduces at the upper heights in the figure and is largely absent at heights above about 95 km. The corresponding zonal winds observed by the radar also display a semi-annual cycle, but the height/time regions of westward winds (negative zonal wind) are rather more extensive than those of the ExUM, with an interval of westward winds being observed to last from Jan-May which is not well reproduced by the model. Further, the maximum monthly-mean observed wind speeds are about double those in the ExUM, with observed wind speeds reaching about $40~\mathrm{ms}^{-1}$ at heights near 90 km in June and $-40~\mathrm{ms}^{-1}$ at heights near 80 km in January. Nevertheless, the ExUM reproduces the general semi-annual pattern of zonal winds.

The corresponding monthly-mean meridional winds in the ExUM at heights below about 95 km display a seasonal pattern with northward winds present from about Nov-May and southward winds at other times. At heights above about 95 km, the meridional winds are southward throughout most of the year. The absolute wind speeds are generally much less than the zonal wind speeds and are mostly less than about $5~\mathrm{ms}^{-1}$, although the strongest meridional winds occur at the upper heights in May/Jun when the southward winds reach about $15~\mathrm{ms}^{-1}$. The observed meridional winds over Ascension Island display a generally similar seasonal variation to that of the ExUM. However, the observed wind speeds are slightly larger throughout most of the year and the region of strongest southward flow in Jun/Jul extends to lower heights than in the ExUM.

In summary, comparing the ExUM and observed winds for Ascension Island, we see that some essential features are well captured and that the semi-annual variation is reproduced. However, there remain some notable differences in detail. This is particularly notable in Feb/Mar when the observed strong westward winds are not well reproduced.

Secondly, we consider the monthly-mean wind fields at the location of Rothera. Here, the ExUM zonal wind is predominantly eastward from Nov-May (i.e, through summer and into autumn) and reverses to be westward from Jul-Oct. The austral summer months exhibit a strong wind shear with velocities increasing from about $-20~\mathrm{ms}^{-1}$ at heights of 80 km to more than $75~\mathrm{ms}^{-1}$

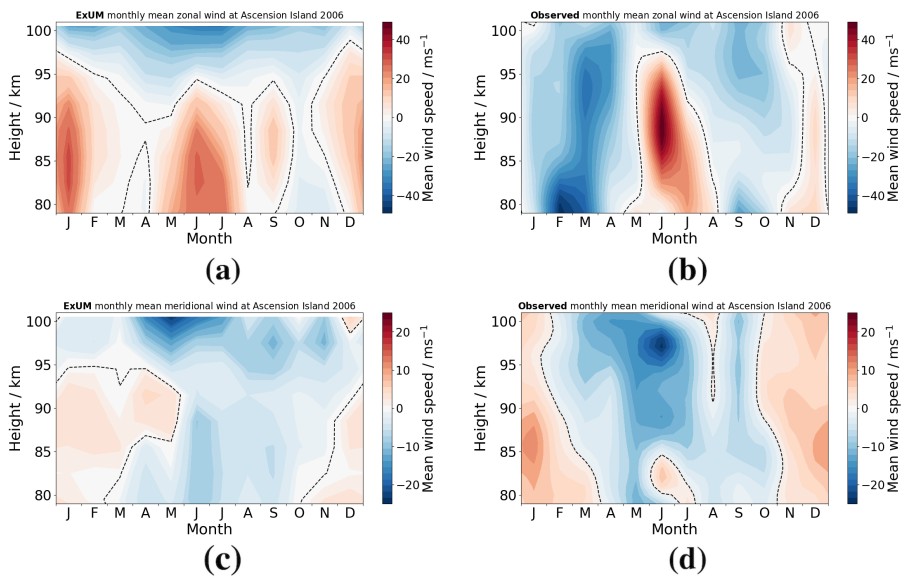

**Figure 6.** Time-height monthly-mean wind contours at Ascension Island in 2006 comparing **(a)** ExUM zonal wind with **(b)** observed zonal wind and **(c)** ExUM meridional wind with **(d)** observed meridional wind. The dashed black line represents the zero-wind line. Colour bars are kept consistent left to right for comparison.

at heights of 100 km. The radar observations from Rothera reveal rather smaller absolute wind speeds in austral summer with values ranging from about -25 ms$^{-1}$ at heights of 80 km to about 25 ms$^{-1}$ at heights of 100 km – significantly less than predicted by the ExUM. The observed winds in winter are noticeably different from those of the ExUM. In particular, the observed winds are eastwards at all heights from Mar-Oct and reach speeds of more than 20 ms$^{-1}$, whereas the ExUM yields westward winds at heights above about 85 km with speeds reaching -20 ms$^{-1}$ for most of these months. This is probably the most notable difference between the winds of the ExUM and those observed by the radars.

The ExUM meridional winds at the location of Rothera exhibit a seasonally reversing pattern with southward flow at all heights in Mar-Jun and regions of northward flow in the other months. In the austral summer months of Nov/Dec, there are southward winds at heights above about 90 km and northwards winds below that height. The radar observations of meridional winds over Rothera reveal a broadly similar pattern of winds to those of the ExUM from Jan-Aug, although with rather stronger northward winds in Jan/Feb. However, in Aug-Dec the observed winds are rather different from those of the ExUM. In particular, the observed winds are almost entirely northward at all heights and actually reach the largest values measured in December, whereas the ExUM winds are actually southwards in Nov/Dec at heights above about 90 km.

In summary, comparing the ExUM and observed winds for Rothera, we see that some aspects of the seasonal variation of the observed winds are reproduced well in the ExUM, particularly below 85 km. However, there is a notable difference in that the observed zonal winds are eastwards in austral winter at all heights whereas in the ExUM they are westwards except at the

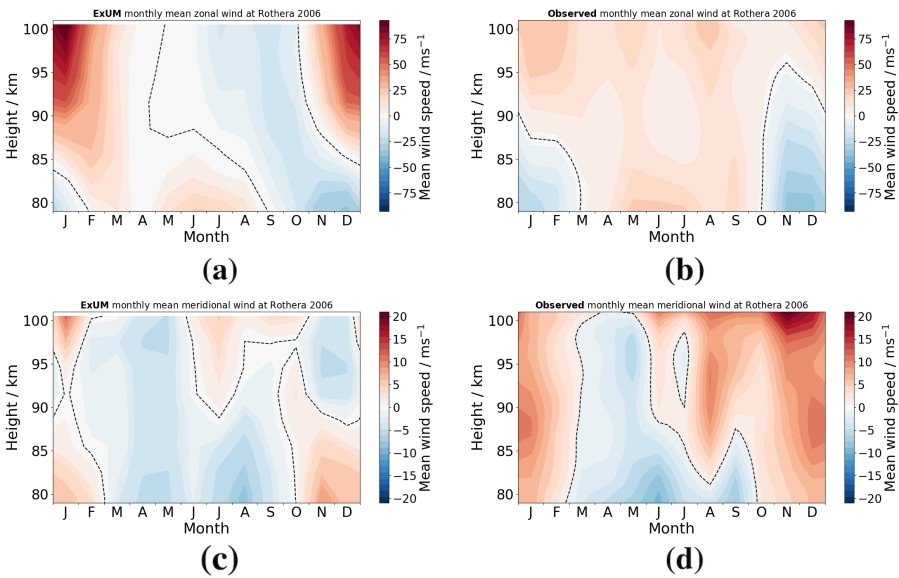

**Figure 7.** Time-height monthly-mean wind contours at Rothera in 2006 comparing **(a)** ExUM zonal wind with **(b)** observed zonal wind and **(c)** ExUM meridional wind with **(d)** observed meridional wind. The dashed black line represents the zero-wind line. Colour bars are kept consistent left to right for comparison.

lowest heights. As well as this, the magnitude of the ExUM winds above 90 km in austral summer is also significantly larger than that observed. We will consider possible explanations for these differences in Sect. 4.

### 3.2 Diurnal tides

We now proceed to a more detailed comparison of the diurnal and semi-diurnal tidal amplitudes and phases in the ExUM at the two locations to those observed by the radars. As with the winds, we will consider monthly-mean properties because they provide a test of the model's ability to reproduce the seasonal variation of the atmosphere.

Monthly-mean tidal amplitudes and phases at heights of 80–100 km were calculated as described in Sect. 2 for both the ExUM results and the radar observations.

#### 3.2.1 Amplitudes

For the location of Ascension Island, the zonal and meridional amplitude components are presented in Fig. 8.

In each panel of the figures, the amplitudes predicted by the ExUM are plotted alongside the meteor radar observations. The shaded regions denote the standard deviation from the curve fitting algorithm, and the black bars indicate the standard deviation from the mean of the measured amplitudes across the month.

Considering the monthly-mean ExUM results, we see that the ExUM tidal amplitudes in most months increase from values of about $10$–$20\,\mathrm{ms^{-1}}$ at heights near 80 km to about $20$-$40\,\mathrm{ms^{-1}}$ at heights of 100 km. However, in Jan and Mar the amplitudes

do not increase across this height range. The zonal and meridional amplitudes are generally similar, but not exactly the same.
For instance, in May and Nov the meridional amplitudes are notably larger than the zonal amplitudes. In fact, the largest amplitudes in the ExUM occur in May when the meridional component amplitude at a height of $100 \mathrm{~km}$ exceeds $50 \mathrm{~ms}^{-1}$.

The corresponding observed tides display a generally similar behaviour, with amplitudes at the lower heights typically being in the range $10$–$20 \mathrm{~ms}^{-1}$ and increasing to larger values at the upper heights, except in Jan and Dec when the amplitudes remain approximately constant with height.

In terms of agreement between ExUM and observed amplitudes, the agreement tends to be better for the zonal components than for the meridional components, and in general the agreement is best at lower altitudes. For the zonal components, excellent agreement is observed in the majority of months. May is the biggest exception, which differs from the observed amplitude by $20$–$30 \mathrm{~ms}^{-1}$ in the worst case. Otherwise, deviations from observed amplitudes are around $10$–$20 \mathrm{~ms}^{-1}$ at most. Looking more closely at their relative magnitudes, the ExUM zonal amplitudes are often greater than or equal to the observed amplitudes up
to $90 \mathrm{~km}$ and then less than or equal to the observed amplitudes above $90 \mathrm{~km}$. For the meridional components, excellent agreement is observed in Jan, Jul, Aug and Oct. The months of Sep, Nov and Dec are reasonable with deviations of around $10$–$20 \mathrm{~ms}^{-1}$. However, the ExUM amplitudes differ notably (by around $30 \mathrm{~ms}^{-1}$) in the other five months. Looking more closely at their relative magnitudes, the amplitudes are similar between the two for Jan, Apr, Jul-Sep and Oct; the ExUM amplitudes are smaller in Feb & Mar, and are larger in May, Jun, Nov & Dec.

Next, we will consider the equivalent monthly-mean diurnal tidal amplitudes at Rothera, which are shown in Fig. 9. Once more, we first consider the ExUM amplitudes. The zonal and meridional components are of similar, small, magnitude for all months. For the majority of months, the amplitudes remain roughly constant with increasing height, however some growth of amplitude with increasing height is observed for Nov-Feb. Maximal amplitudes of c. $20 \mathrm{~ms}^{-1}$ are seen in January for both the zonal and meridional components.

Secondly, we consider the observed amplitudes. These also have zonal and meridional components which are of similar, small, magnitude for all months. The amplitude remains roughly constant with increasing height in all cases. Maximal amplitudes of c. $15 \mathrm{~ms}^{-1}$ and c. $20 \mathrm{~ms}^{-1}$ are observed for the November zonal component and the December meridional component respectively.

In terms of agreement between ExUM and observed amplitudes, the agreement is excellent across all months for both zonal
and meridional components. The magnitudes are similar for both components for all months.

### 3.2.2 Phases

The tidal phases are defined as the Local Time at which the tidal wind first reaches a maximum value for a particular component. Phases were calculated for zonal and meridional components for both the ExUM and observed winds at both Ascension Island and Rothera. As with the amplitudes, we present figures on which we plot both the ExUM tidal phases and the observed tidal
phases.

The monthly diurnal tidal phases are presented for both the zonal and meridional wind components at Ascension Island in Fig. 10.

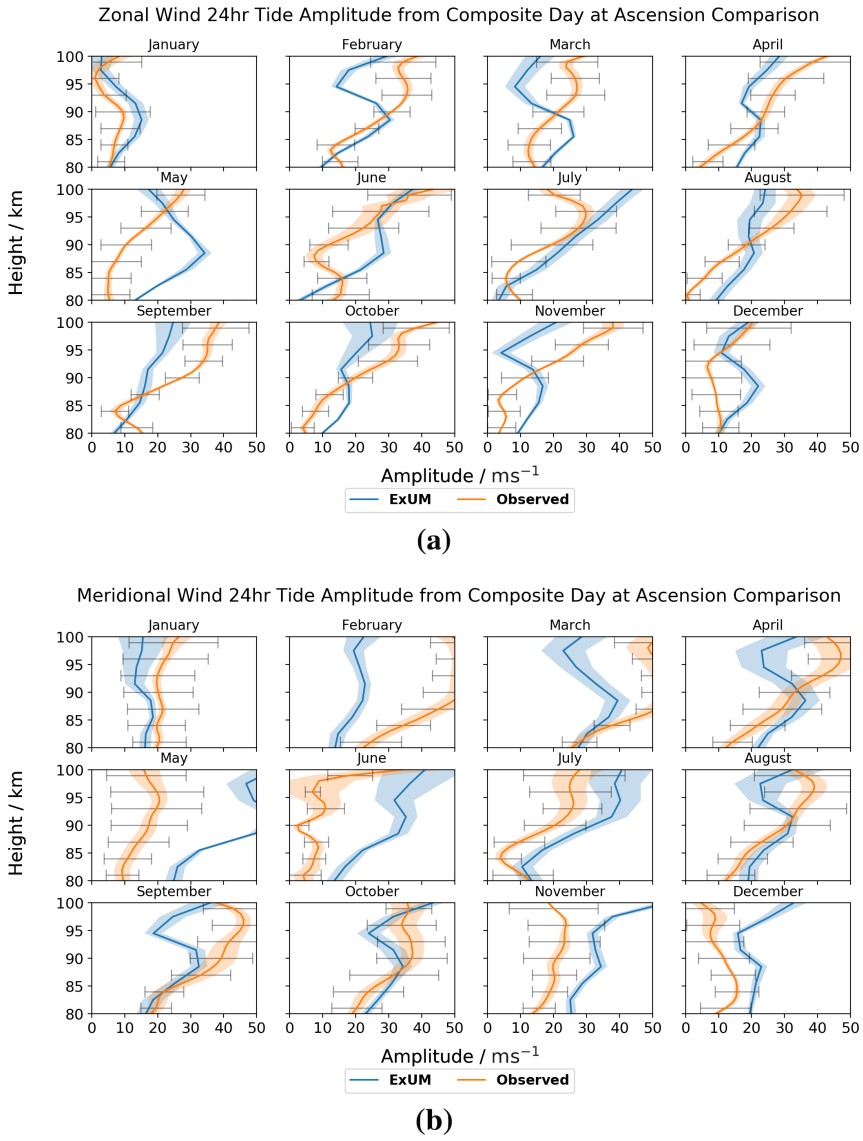

**Figure 8.** Amplitudes for each month as a function of height for the **(a)** zonal ($u$) and **(b)** meridional ($v$) component of the diurnal tide at Ascension Island. The shaded regions denote the standard deviation from the curve fitting algorithm, and the black bars indicate the standard deviation from the mean of the measured amplitudes across the month. Both the amplitudes from the model (blue) and meteor radar (orange) are plotted.

Firstly, we consider the ExUM phases. The ExUM meridional phases are consistent in leading their zonal counterparts, by around 4–8 h. A smooth decrease in phase with increasing height is observed for the majority of months indicative of upwardly

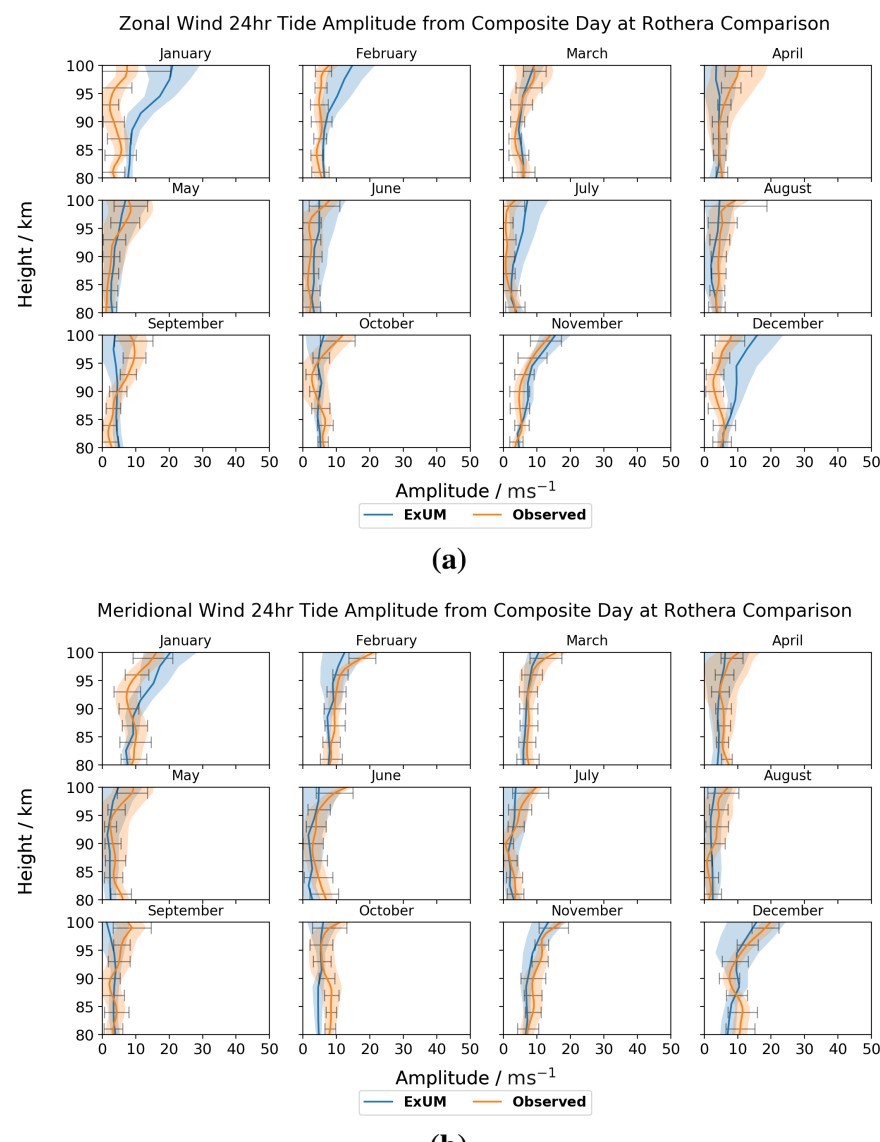

**Figure 9.** Amplitudes for each month as a function of height for the **(a)** zonal ($u$) and **(b)** meridional ($v$) component of the diurnal tide at Rothera. The shaded regions denote the standard deviation from the curve fitting algorithm, and the black bars indicate the standard deviation from the mean of the measured amplitudes across the month. Both the amplitudes from the model (blue) and meteor radar (orange) are plotted.

propagating tides. In the zonal component, a lesser decrease with increasing height (namely a steeper phase gradient) is seen at high altitudes in Aug-Nov.

Secondly, we consider the observed phases. As with the modelled phases, the observed meridional phases consistently lead the observed zonal phases by around 2–8 h, with June being the only exception where the observed meridional component is anomalous. This is to be expected given the lack of observations in June (see Fig. 4). June is also the only exception to a smooth decrease in phase with increasing height in the meridional component. In the zonal component, a decrease in phase with increasing height is observed in the majority of months, with the exceptions being at lower altitudes.

In terms of agreement between ExUM and observed phases, qualitative agreement in the characteristics of the phases is in general excellent across both components in all months. In terms of a quantitative comparison, for the zonal component, in several cases the ExUM phase is in excellent agreement, such as in Jan, May, Jun and Nov. However, in other months the observed phase leads the ExUM phase by 4–8 h, and the observed phases often have a steeper slope with height indicative of longer vertical wavelengths. For the meridional component, the observed phase leads the ExUM phase by 4–10 h in all months, but the phase slope is similar between ExUM and observed phases for all months.

Next, the monthly diurnal tidal phases are presented for both the zonal and meridional wind components at Rothera in Fig. 11. It should be noted that the amplitudes for many months are small, and so caution must be taken in drawing conclusions from the corresponding phases. Nevertheless, we can look for qualitative features.

Again, we firstly consider the ExUM phases. The ExUM meridional phases are once more consistent in leading their zonal counterparts. In both components the phases remain roughly constant with increasing height for the majority of the year.

Secondly, we consider the observed phases. As with the modelled phases, the observed meridional phases consistently lead the observed zonal phases, with July being the only exception. In the zonal component, a general trend of decrease in phase with increasing height is seen in the majority of months, with the exceptions being at higher altitudes. In the meridional component, for most months the phase is roughly constant with increasing height, with a weak decrease in phase with increasing height observed in some months. July is again the exception where an increase in phase with increasing height is observed for lower altitudes.

In terms of agreement between ExUM and observed phases, the agreement is better for the meridional component which is on the whole very good. For the zonal component, in several cases the ExUM phase is in excellent agreement, such as in Feb, Mar and Oct-Dec. However, in other months the ExUM phase leads the observed phase by 4–10 h, and the observed phases in general have a decreasing slope with increasing height which is not apparent in the ExUM phases which on the whole have constant phase slope. For the meridional component, the agreement is in general excellent across the majority of months, with the roughly constant phase slope mirrored. The agreement is worse for May-Jul where the ExUM phase leads the observed phase by up to 10 h.

To summarise, this first comparison is indicative of the ExUM's strong ability to capture the diurnal tidal phases and amplitudes, with order of magnitude and qualitative agreement across many of the diagnostics considered, with no specific tuning necessary. Core qualitative features are reproduced – large amplitudes at Ascension Island compared with small amplitudes at Rothera; a general increase in amplitude with height; a general decrease in phase with height; the meridional tide component exceeding the zonal component; and the meridional phases leading their zonal counterparts.

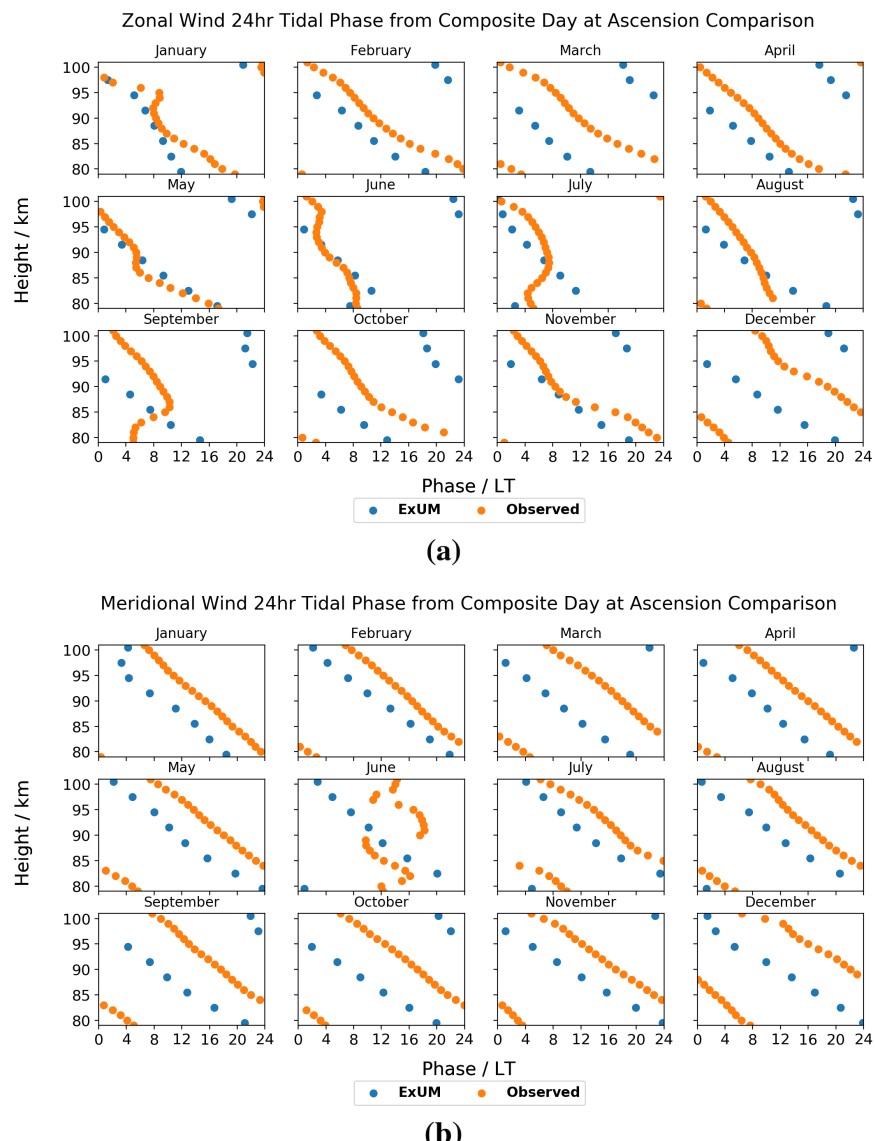

**Figure 10.** Phases for each month as a function of height for the **(a)** zonal ($u$) and **(b)** meridional ($v$) component of the diurnal tide at Ascension Island. Both the phases from the model (blue) and meteor radar (orange) are plotted.

The differences observed in amplitude do not follow a clear trend, but often the accuracy of the amplitudes in comparison with observed values is in general better at lower altitudes, and more differences were seen towards the upper heights of the model. The modelled phases systematically lead the observed phases by around 4–10 h. Where differences in phase gradient are evident, at Ascension Island, the observed phase gradients are often steeper than that seen in the ExUM and at Rothera, the
ExUM phase gradients are generally vertical, which is not always the case in observations.

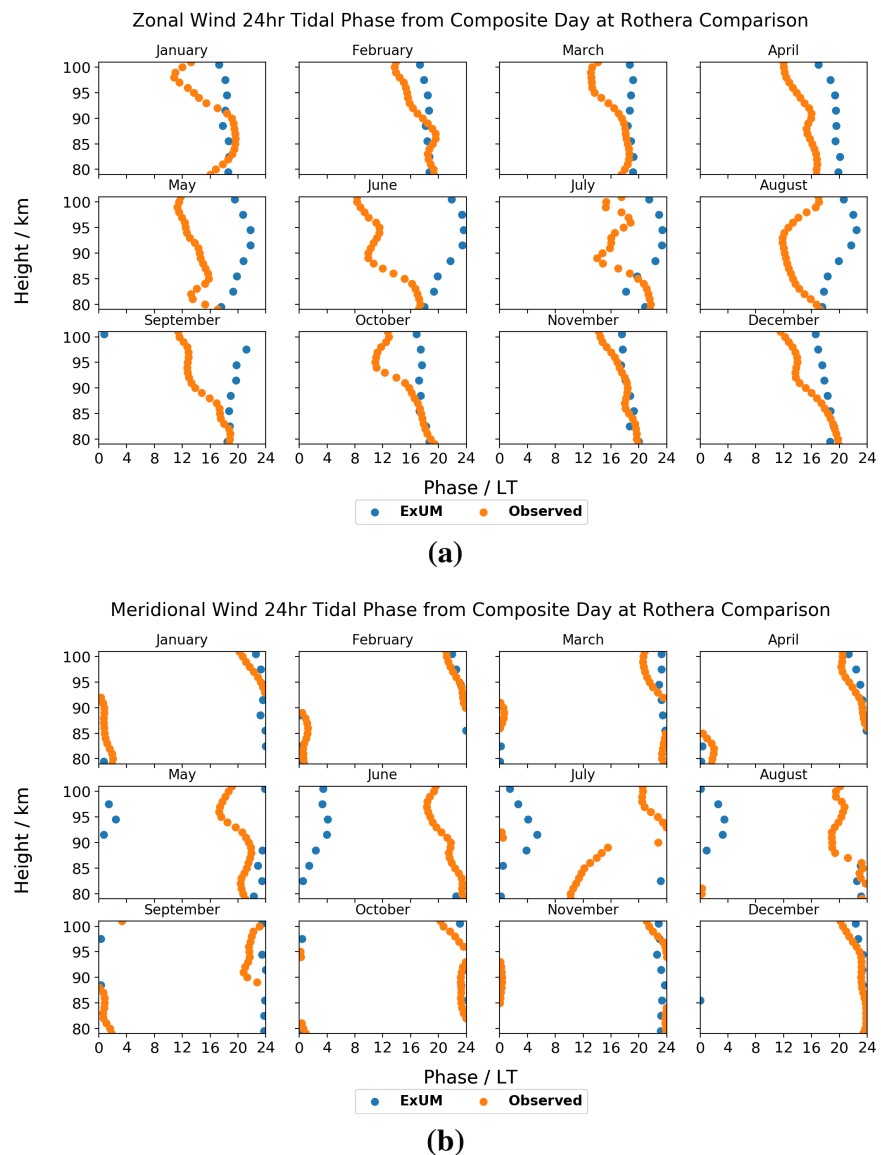

**Figure 11.** Phases for each month as a function of height for the **(a)** zonal ($u$) and **(b)** meridional ($v$) component of the diurnal tide at Rothera. Both the phases from the model (blue) and meteor radar (orange) are plotted.

## 3.3 Semi-diurnal tides

We now proceed to a detailed comparison of tidal amplitudes and phases for the semi-diurnal tide, from both the ExUM and meteor radar observations at both locations.

Monthly-mean tidal amplitudes and phases at heights of 80–100 km were calculated as described in Sect. 2 for both the
ExUM results and the radar observations.

### 3.3.1 Amplitudes

In each panel of the figures, the amplitudes predicted by the ExUM are plotted alongside the meteor radar observations. The
shaded regions denote the standard deviation from the curve fitting algorithm, and the black bars indicate the standard deviation
from the mean of the measured amplitudes across the month.

The monthly semi-diurnal tidal amplitudes are presented for the zonal and meridional wind components at Ascension Island
in Fig. 12.

Firstly, we consider the ExUM amplitudes. The ExUM meridional amplitudes are, for the majority of months, greater than or
equal to the corresponding zonal amplitudes, particularly above 90 km. October does not fit this trend; the zonal component is
larger than the meridional component, again more so above 90 km. Growth of amplitude with increasing height is observed for
the majority of months in the ExUM meridional amplitudes, and the amplitudes remain roughly constant with increasing height
for the ExUM zonal amplitudes. The months which do not follow this pattern are Sep-Dec, where the opposite is true; namely
the zonal amplitudes grow with increasing height whereas the meridional amplitudes remain roughly constant with increasing
height. The largest amplitudes of c. 59 $\mathrm{ms}^{-1}$ are observed when looking more closely at the meridional components, in June.

Secondly, we consider the observed amplitudes. We note that these amplitudes have meridional components which are
greater than or equal to their zonal counterparts. In particular, Oct-Dec have similar amplitudes in both components. The
observed zonal amplitudes on the whole remain constant with increasing height, with a slight increase evident in Jan, Jun
and Aug. The meridional amplitudes remain roughly constant with increasing height for May and Oct-Feb, and grow with
increasing height for Mar, Apr and Jun-Aug. The largest amplitudes of c. 42 $\mathrm{ms}^{-1}$ are observed when looking more closely at
the meridional components, in June.

In terms of agreement between ExUM and observed amplitudes, the agreement is excellent and is marginally better for
the zonal components in comparison with the meridional components and in general is best at lower altitudes. For the zonal
components, excellent agreement is observed in the majority of months. October is the biggest exception, which differs from
the observed amplitude by 20–30 $\mathrm{ms}^{-1}$ at 100 km. Otherwise, deviations from observed amplitudes are around 5–15 $\mathrm{ms}^{-1}$.
Looking more closely at their relative magnitudes, the ExUM zonal amplitudes are similar to observed amplitudes in the
majority of cases, but tend to be larger where the amplitudes do differ. For the meridional components, excellent agreement is
observed once more in the majority of months. Jan and Aug are the main exceptions, with deviations of around 20 $\mathrm{ms}^{-1}$ at
higher altitudes, but still show excellent agreement below 90 km. Otherwise the difference is minimal at 5–10 $\mathrm{ms}^{-1}$. Looking
more closely at their relative magnitudes, the amplitudes are once more similar between the two, with the ExUM amplitudes
again larger where they differ.

Next, the monthly semi-diurnal tidal amplitudes are presented for the zonal and meridional wind components at Rothera in
Fig. 13.

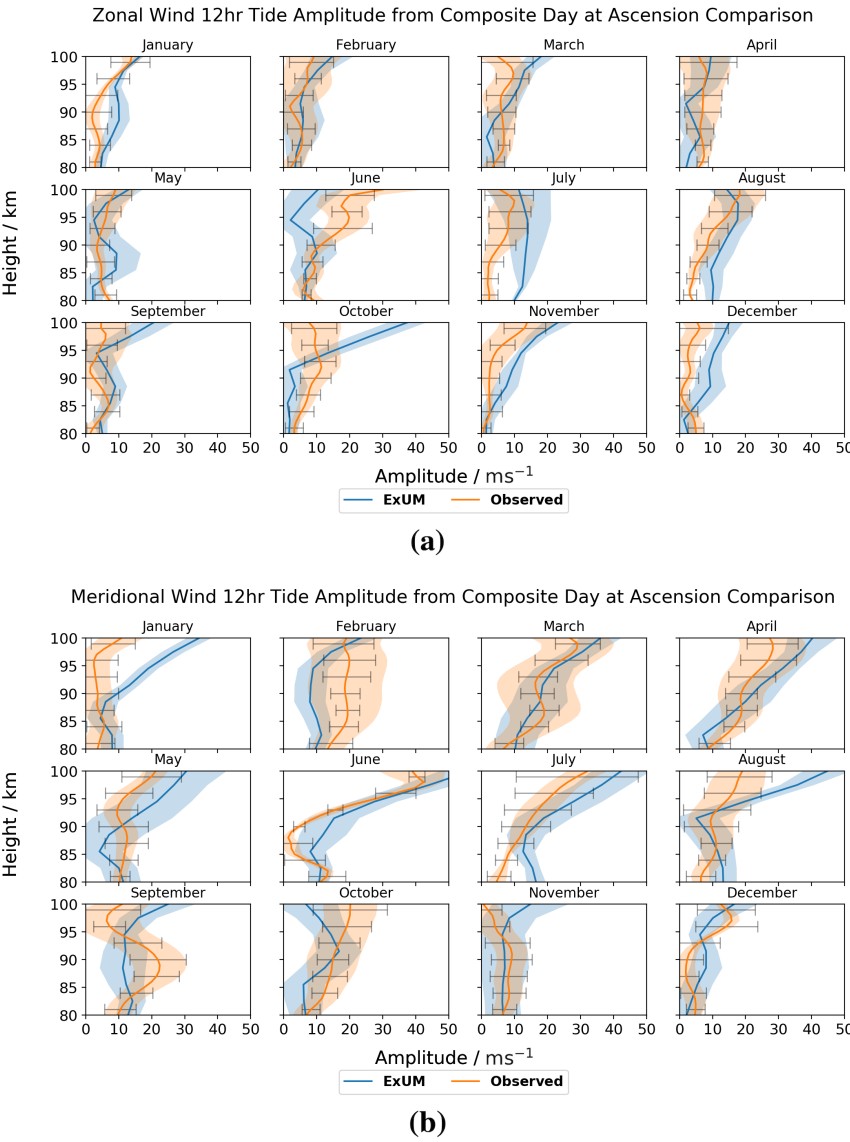

**Figure 12.** Amplitudes for each month as a function of height for the **(a)** zonal ($u$) and **(b)** meridional ($v$) component of the semi-diurnal tide at Ascension Island. The shaded regions denote the standard deviation from the curve fitting algorithm, and the black bars indicate the standard deviation from the mean of the measured amplitudes across the month. Both the amplitudes from the model (blue) and meteor radar (orange) are plotted.

Once more, we first consider the ExUM amplitudes. The ExUM amplitudes are of very similar magnitude across both components – therefore we will summarise them both simultaneously. The growth of amplitude with height is evident across nearly all months, with March being the only exception. We observe the largest amplitudes of c. $40 \, \mathrm{ms}^{-1}$ in Dec/Jan.

Secondly, we consider the observed amplitudes. The observed amplitudes are also of very similar magnitude across both components. The amplitudes exhibit growth with increasing height in Mar-May and less clearly so in Aug-Oct. For the remaining six months, the amplitude remains roughly constant with increasing height. The largest amplitudes of c. 35 $\mathrm{ms}^{-1}$ are apparent in April.

In terms of agreement between ExUM and observed amplitudes, the agreement is mirrored for both the zonal and meridional components. Excellent agreement is observed in the majority of months with the best agreement in general at lower altitudes. Dec-Mar show the largest deviations of around 10–20 $\mathrm{ms}^{-1}$ at higher altitudes. Otherwise, the agreement is excellent with deviations of around 5–10 $\mathrm{ms}^{-1}$. Looking more closely at their relative magnitudes, in general the amplitudes are similar. In the few cases they do differ, no obvious trend is apparent – for some months the ExUM amplitudes are larger and for others they are smaller.

### 3.3.2  Phases

Along with the amplitudes, the monthly semi-diurnal tidal phases (namely, the hours of peak amplitude) were also calculated for zonal and meridional components for both the ExUM and observed winds at both Ascension Island and Rothera. Once more the phases predicted by the ExUM are plotted alongside those predicted by the meteor radar observations.

The monthly semi-diurnal tidal phases are presented for both the zonal and meridional wind components at Ascension Island in Fig. 14. It should be noted that the amplitudes for many months are small, and so caution must be taken in drawing conclusions from the corresponding phases. Nevertheless, we can as before look for qualitative features.

Firstly, we consider the ExUM phases. The ExUM meridional phases in general lead their zonal counterparts. A decrease in phase with increasing height is observed for the majority of months indicative of upwardly propagating tides, however it is worth noting that the corresponding phase gradient is much shallower than that seen for the phases of the diurnal tide, and thus indicative of a shorter vertical wavelength. In the zonal component in Mar, Nov and Dec, the phase becomes roughly constant with increasing height at high altitudes, and May shows an increase in phase with increasing height also above 90 km.

Secondly, we consider the observed phases. Contrary to the modelled phases, it is not at all obvious that there is a trend between the observed zonal and meridional phases. In general, the trend of decrease in phase with increasing height is apparent in the majority of months for both components. However, other trends are observed. In the zonal component, the months of Mar, May, Jun and Oct all have heights for which the phase remains roughly constant with increasing height. Jan, May, Sep and Nov exhibit an increase in phase with increasing height at various heights. In the meridional component, there are fewer exceptions to the general trend of a decrease in phase with increasing height. May/Jun have periods of constant phase with increasing height at around 95 km; and September exhibits an increase in phase with increasing height also around 95 km.

In terms of agreement between ExUM and observed phases, the agreement is in general better for the meridional component which is excellent for many months, such as Jan-Apr, Oct and Nov, but has larger differences in Jul-Aug. It is interesting to note that the model matches some of the less expected behaviour such as the increase in phase with increasing height above 90 km in September, however the observed amplitudes are fairly small here. For the zonal component, the agreement tends to be good at best, in months such as Jan, Sep and Oct. Again it is interesting that some more complex features are well captured

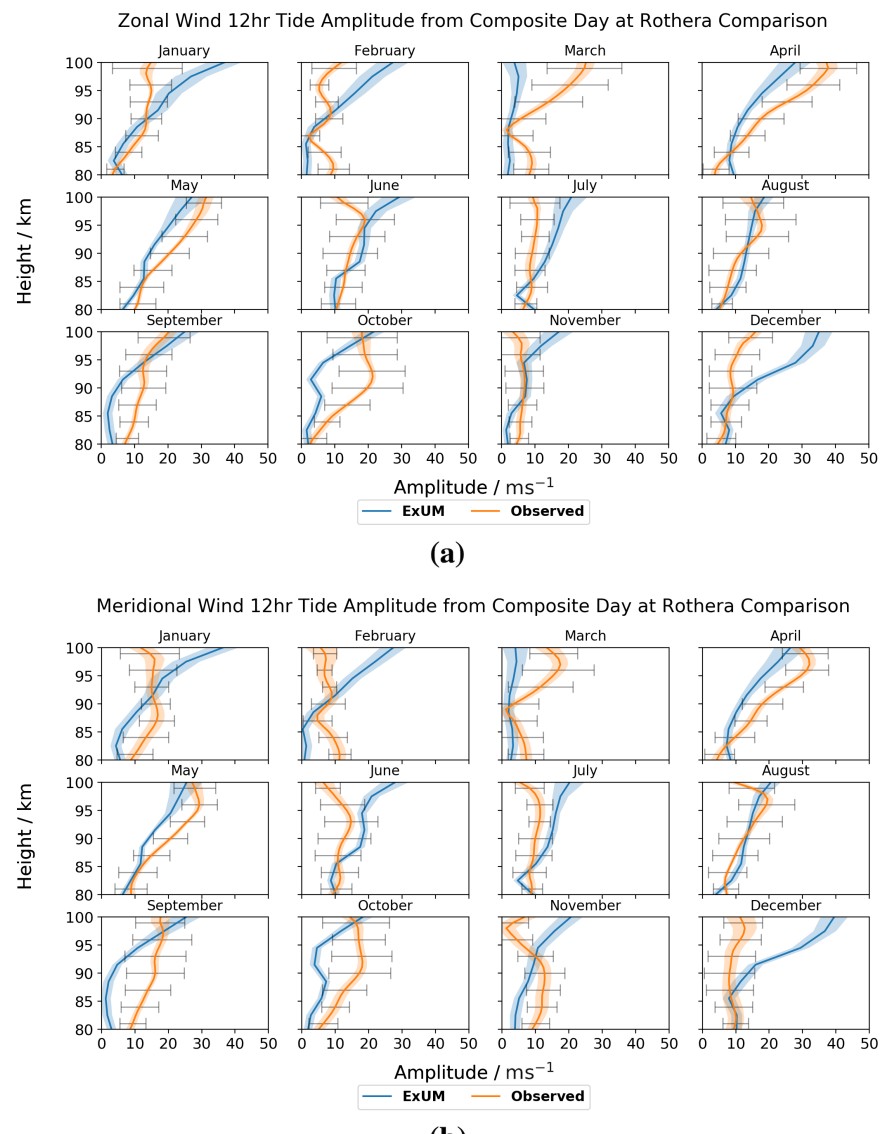

**Figure 13.** Amplitudes for each month as a function of height for the **(a)** zonal ($u$) and **(b)** meridional ($v$) component of the semi-diurnal tide at Rothera. The shaded regions denote the standard deviation from the curve fitting algorithm, and the black bars indicate the standard deviation from the mean of the measured amplitudes across the month. Both the amplitudes from the model (blue) and meteor radar (orange) are plotted.

in September. The roughly constant phase with increasing height is also captured in March, but is out of phase by around 2–3 h. This characteristic is repeated in many other months, such as Jul-Aug; namely, the correct qualitative behaviour is seen, but they are out of phase by 2–6 h. However, in other months the ExUM phases do not match those observed, in particular in Nov

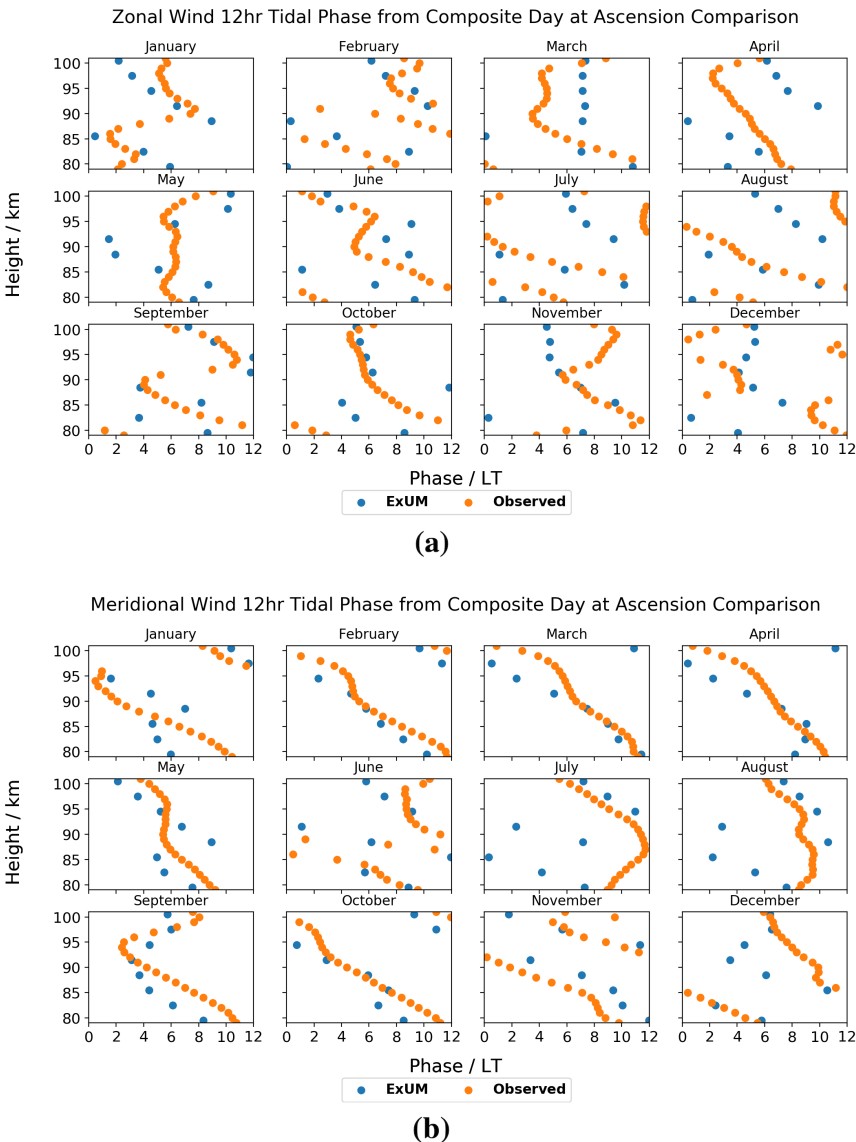

**Figure 14.** Phases for each month as a function of height for the **(a)** zonal ($u$) and **(b)** meridional ($v$) component of the semi-diurnal tide at Ascension Island. Both the phases from the model (blue) and meteor radar (orange) are plotted.

and Dec. In general though, the agreement is good and the trend of a more shallow decrease in phase with increasing height is mirrored between the ExUM and observed phases.

Next, the monthly semi-diurnal tidal phases are presented for the zonal and meridional wind components at Rothera in Fig. 505 15.

Again, we firstly consider the ExUM phases. The ExUM meridional phases are once more consistent in leading their zonal counterparts, by around 2–6 h. Apart from this phase shift, the zonal and meridional components are practically identical across all months. The phases exhibit a general trend of decrease with increasing height. This decrease is steeper in some months than others, for example compare Feb/Mar (where it is shallow) with Sep/Nov (where it is steeper). This is indicative of varying vertical wavelength throughout the year, but of consistently upwardly propagating tides.

Secondly, we consider the observed phases. As with the modelled phases, the observed meridional phases consistently lead the observed zonal phases by around 3–6 h. They also share the property that, apart from this phase shift, the zonal and meridional components are very similar across the majority of months. A general trend of decrease in the observed phase with increasing height is seen. The observed phases also exhibit a variety of phase gradients, with shallower gradients in Mar and Oct, and steeper gradients in Jun-Sep, for example.

In terms of agreement between ExUM and observed phases, the agreement on the whole is very good, and is marginally better for the zonal component. The amount that the meridional component leads the zonal component is on the whole consistent across both phases. For the zonal component, in several cases the ExUM phase is in excellent agreement, such as in Jan, Mar, Jun, Aug and Sep, with a reasonably consistent phase gradient between the two. For several months, the main difference is that the observed phases have a steeper slope with increasing height (such as in May-Jul), indicative of longer vertical wavelengths. For the meridional component, it is a similar story in comparison with the zonal component; Jan, Mar, Jun, Aug and Sept show excellent agreement on the whole and a steeper phase slope is seen in the observed phases in May-Jul. The agreement is however slightly worse between 80 and 90 km in Mar and Sep. In both components, Feb and Nov are less similar, and show several qualitative differences. In general though, agreement is good with little phase shift between ExUM and observed phases with general trends such as the decrease in phase with increasing height well captured.

To summarise, the ExUM results capture some of the characteristic features of the observed diurnal and semi-diurnal tidal amplitudes and phases across many of the diagnostics considered, with no specific tuning beforehand. Key qualitative features are reproduced, including large diurnal amplitudes at Ascension Island (particularly in the zonal component) and large semi-diurnal amplitudes at Rothera, a general increase in amplitude with height, a general decrease in phase with height (indicating upward propagation), a similar magnitude for zonal and meridional components and meridional phases that lead their zonal counterparts.

In the particular case of the semi-diurnal amplitudes, notable differences between the ExUM and the radar observations are often more pronounced at the greater heights.

Finally, the ExUM semi-diurnal phases systematically lead the observed phases by around 2–6 h at Ascension Island. Where differences in phase gradient are evident, the observed phase gradients are often slightly steeper than those seen in the ExUM at both locations, indicating that tidal vertical wavelengths in the ExUM are slightly shorter than observed.

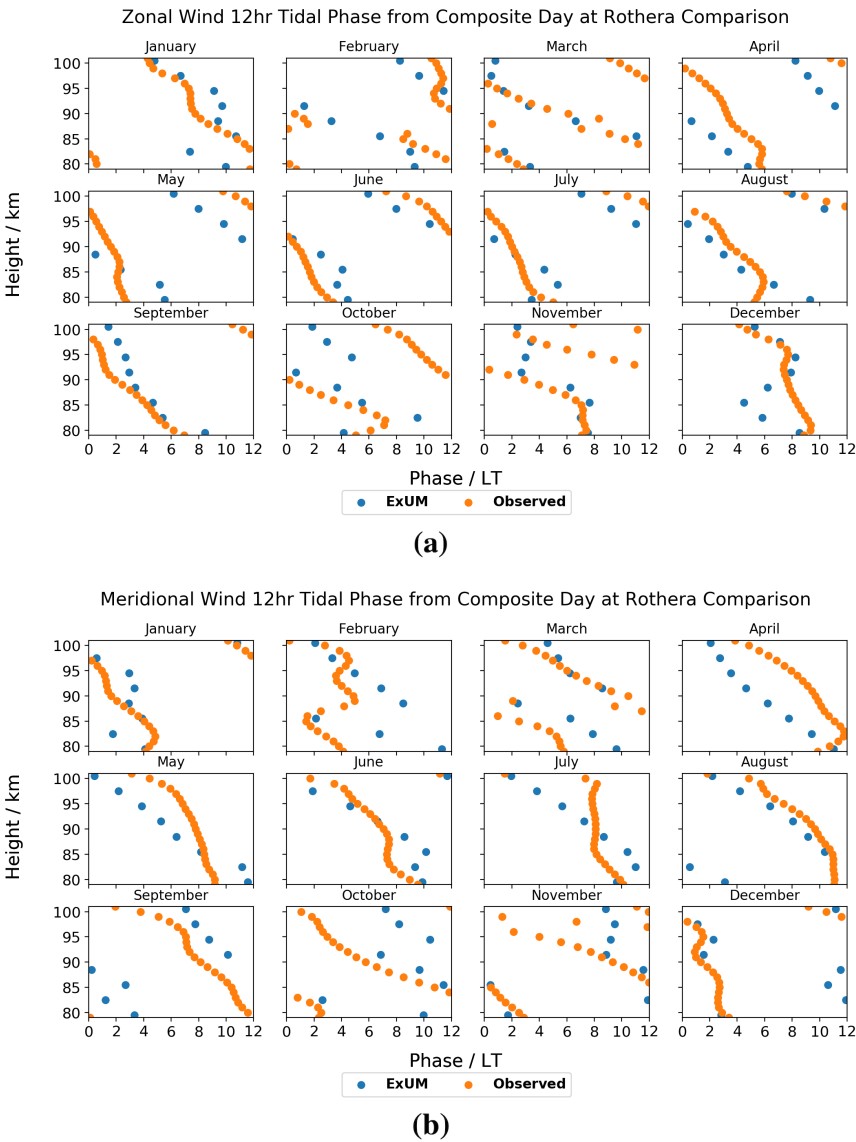

**Figure 15.** Phases for each month as a function of height for the **(a)** zonal ($u$) and **(b)** meridional ($v$) component of the semi-diurnal tide at Rothera. Both the phases from the model (blue) and meteor radar (orange) are plotted.

## 4 Discussion

The results presented above reveal that there are many aspects of the background winds and the diurnal and semi-diurnal tides in the ExUM that agree well with observations made in the MLT by the meteor radars at the two sites. However, there are also a number of notable differences, or biases. Here we will discuss the possible origins of these biases and consider how the ExUM

might be developed in future to reduce them. Note that the focus of our discussion will be on the ExUM's representation of background winds and tides and how they compare to the observations. More complete investigations of the observed winds and tides themselves over these locations and discussions of how they compare to other observational studies can be found in Davis et al. (2013) for Ascension Island and Sandford et al. (2010) and Dempsey et al. (2021) for Rothera.

## 4.1 Monthly mean winds

The most striking difference between the ExUM's monthly-mean zonal and meridional winds in the MLT and those observed by the radars occur in two places, i) the Antarctic during austral summer, when the ExUM zonal winds at the upper heights are much stronger than observed over Rothera, and ii) the austral winter, when the observations reveal eastward winds at all heights from March to October, but the ExUM predicts westward winds commencing in April, i.e, the observed winds are actually in the opposite direction to those predicted by the ExUM.

The first of these differences most likely arises from the gravity-wave parametrizations used in the ExUM, which are not yet tuned for the high-latitude MLT and so may give rise to unrealistically high mean-flow accelerations. However, the second difference is particularly striking because the existence of any eastwards winds in the polar winter MLT is unexpected since the strong eastwards winds of the underlying winter stratosphere will have removed (by critical-level filtering) all ascending GWs with eastwards phase velocities and momentum flux – leaving no such waves to dissipate in the MLT where they could force eastwards winds.

Recently, an explanation for the existence of such eastwards winds in the polar winter MLT has been proposed in the modelling study of Becker and Vadas (2018). These authors suggest that non-primary GWs are generated in situ over the Southern Andes in winter, either by nonlinear instabilities and/or by the local body forces from the temporally and spatially localized wave drag resulting from the breaking of large-amplitude mountain (orographic) GWs. These non-primary GWs may include waves which have significant eastwards momentum fluxes and which are excited at heights above levels where they would otherwise be removed by the critical-level filtering of eastwards winds. When such eastward waves reach the MLT and themselves dissipate, their eastward momentum may then force eastward mean winds. However, Becker and Vadas (2018) did not have available zonal wind measurements from the austral winter MLT and so could not investigate further. In this context we also note that the MLT winds over Rothera for 2005–2009 reported by Sandford et al. (2010) also included eastward winds in winter. Further, the recent study by Stober et al. (2021) which considered the observations from six high southern latitude radars for the year 2019 also reported wintertime eastward winds over all stations. The results we have presented here suggest that their predicted eastward winds do indeed occur and so our observations are not in disagreement with the work of Becker and Vadas (2018) and suggest that non-primary gravity-waves may play a key role in the circulation of the Antarctic MLT (cf. Becker and Vadas, 2020).

The ExUM, in common with nearly all GCMs, does not include gravity-wave sources above the troposphere and so cannot produce an eastward forcing of the polar winter MLT since any eastward propagating waves in the model will be filtered out by critical levels before reaching the MLT. Therefore, the ExUM cannot produce the observed eastward winds. This limitation

may well explain the lack of eastward polar winter winds also found in other GCMs which launch gravity-waves from the
surface only, including WACCM-X, eCMAM, MUAM and other high atmosphere models.

This bias in the ExUM indicates that further work is required on the GW forcing and parameterization for the MLT, with particular reference to in-situ GW and non-primary GW generation (e.g., Becker and Vadas, 2018, 2020). In this context, it is worth noting that the non-orographic Ultra Simple Spectral Parameterization (USSP) (Warner and McIntyre, 2001) used in the ExUM was designed, and primarily tuned, to obtain more realistic stratospheric features such as the QBO rather than to give
appropriate forcing in the MLT (Scaife et al., 2002).

The USSP scheme treats non-orographic gravity waves with non-zero phase speeds which are unable to be resolved by the model. This is important as the model has too coarse a resolution to represent large portions of the gravity wave spectrum. The approach used is that of Warner and McIntyre (2001) with further modifications (Scaife et al., 2002) to launch an unsaturated spectrum from a level close to the surface and to impose a homogeneous (location invariant) total vertical flux of horizontal
wave pseudomomentum. The spectrum uses a characteristic vertical wavelength peak of 4.3 km and parameterizes vertical wavelengths up to a maximum of 20 km. The amplitude of the spectrum is chosen to give momentum deposition and, hence, a Quasi-Biennial Oscillation (QBO) in the model that is realistic. For comparison with other parameterizations, a typical value of the total launch flux in all four directions is $6.6 \times 10^{-3} \, \mathrm{kg m^{-1} s^{-1}}$.

The scheme also includes the frictional heating due to gravity wave dissipation, and consequent loss of kinetic energy
(see Sect. 3.5 of Walters et al. (2017) for more details), but does not include ionospheric heating effects such as ion drag. The inclusion of GW heating is important as previous studies, for example, by Medvedev and Klaassen (2003), Yiğit and Medvedev (2009) and Hickey et al. (2011) have shown that GWs produce localized, and occasionally very strong, heating and cooling, which certainly plays an important role in the MLT. However, the scheme does not have a latitudinally varying GW spectrum. Yiğit et al. (2021) showed that implementing this type of scheme can have a significant impact on middle atmosphere
circulation, which can therefore have an important effect on the diurnal tides. Therefore, this addition will be a priority in future development of the USSP for the MLT.

To further investigate and demonstrate the role of GWs in forcing the winds of the MLT in the ExUM, we examined the time series of monthly-mean zonal and meridional gravity-wave tendencies from the spectral scheme over the course of 2006. This is presented in Fig. 16.

It is evident from the tendencies in the figure that, as expected, the spectral GW scheme is the dominant driver of the MLT winds. This highlights the need for improvements and modifications in the scheme in the MLT if it is to produce the observed winds. We tested the impact of the USSP on the ExUM MLT winds by simply turning off the USSP scheme. With the USSP off, we attain the monthly mean background winds at Rothera as shown in Fig. 17.

It can be seen from the figure that the ExUM winds with the USSP turned off now more closely resemble those observed over
Rothera in austral summer (Jan-Feb and Nov-Dec). However, the austral winter worsens in comparison which illustrates that a spectral gravity-wave scheme is certainly necessary for the MLT with this horizontal resolution. This highlights the limitations of the GWs parameterized by the USSP and suggests that improvements to the parameterization scheme are necessary to cope

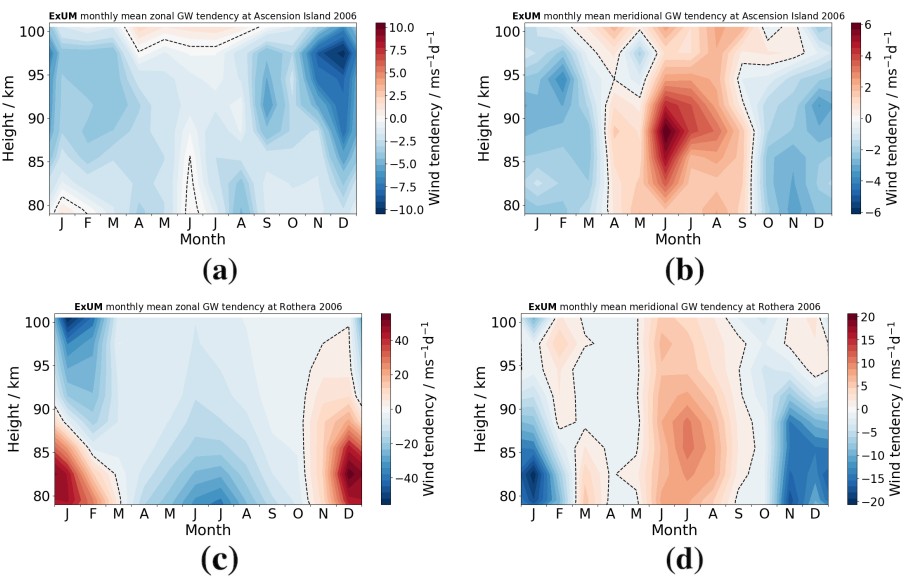

**Figure 16.** ExUM time-height gravity wave tendency contours in 2006. At Ascension Island for the **(a)** zonal ($u$) and **(b)** meridional ($v$) components. Similarly at Rothera for the **(c)** zonal ($u$) and **(d)** meridional ($v$) components.

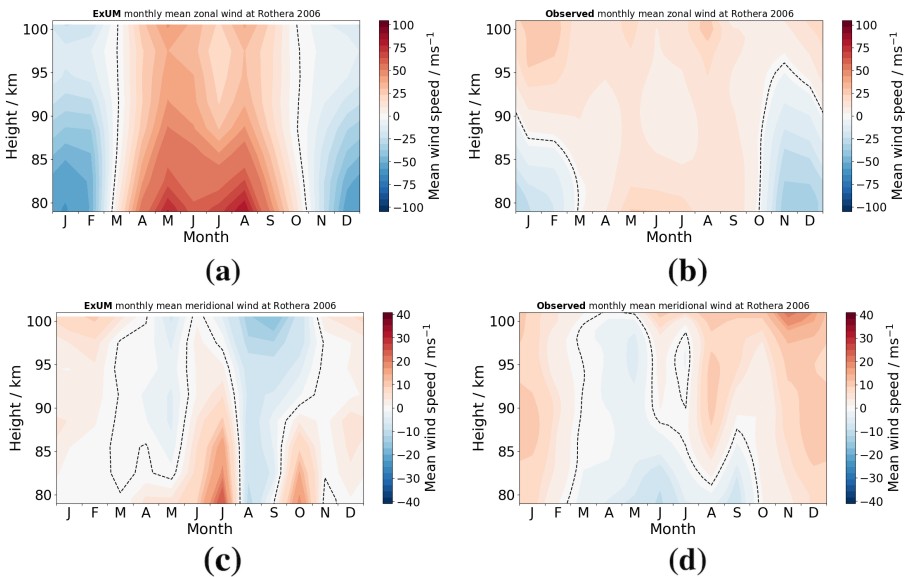

**Figure 17.** Time-height monthly-mean wind contours at Rothera in 2006 comparing **(a)** ExUM zonal wind without the USSP with **(b)** observed zonal wind and **(c)** ExUM meridional wind without the USSP with **(d)** observed meridional wind. The dashed black line represents the zero-wind line. Colour bars are kept consistent left to right for comparison.

with GWs in the MLT – with particular focus on in-situ gravity-wave generation and non-primary (e.g. secondary) GWs which have been shown to also give the observed eastward winds (Becker and Vadas, 2018).

We also place these results in the context of the recent publication of Miyoshi and Yiğit (2019). They used the Kyushu GCM and incorporated the nonlinear spectral GW scheme of Yiğit et al. (2008). They showed that the non-orographic subgrid-scale GWs attenuate the migrating semi-diurnal solar-tide (SW2) amplitude for solstice conditions in the lower thermosphere and modify the latitudinal structure of the SW2 above a 150 km height. On inspection of the amplitudes of the semi-diurnal tide produced by the model with the USSP off at the two locations considered (not shown), it is also clear that the USSP acts to attenuate the semi-diurnal tidal amplitudes under solstice conditions, typically by around $20 \text{ ms}^{-1}$ at a height of $100 \text{ km}$ (primarily in Jun - Sep).

## 4.2 Diurnal and semi-diurnal tides

The results presented above for the tides show that the ExUM captures many of the main features of both diurnal and semi-diurnal tides at Ascension Island and Rothera. However, the semi-diurnal tide at Ascension Island and the diurnal tide at Rothera reach only small amplitudes in both the ExUM and the observations and so the model biases may not be meaningful. We will therefore restrict our discussion to the larger amplitude tides that dominate the motion field at each location – that is, the diurnal tide at Ascension Island and the semi-diurnal tide at Rothera.

In the case of the diurnal tide over Ascension Island, the ExUM tidal amplitudes are in most months in good agreement with the observations and increase with height in a manner similar to that observed. However, there are differences in amplitude of greater than $20 \text{ ms}^{-1}$ at some heights in some months in one or both components. This is particularly apparent in February, May and June in the meridional component.

In the case of the semi-diurnal tide over Rothera, the ExUM amplitudes are again generally in reasonable agreement with those observed, but there are some months where the ExUM amplitudes are rather larger than observed (January and December at the upper heights) or smaller than observed (September and October at the lower heights).

At Ascension Island, the diurnal tidal phases have gradients (vertical wavelengths) that are in excellent agreement with the observations, although the absolute values of phase in the ExUM in most months lead the observed phases by about 3–4 h. This systematic difference may, in part, reflect the accumulated phase difference over several cycles of the (short vertical wavelength) tide as it propagates from its sources at lower heights if there is a mismatch between the model vertical wavelength and that of the tide in the real atmosphere.

In the case of the semi-diurnal tide at Rothera, the phases are less well defined than is the case at Ascension Island. Indeed, in some months the vertical profile of tidal phase has a complicated structure without a uniform gradient across the height range considered. This is evident in both the ExUM results and the observations and is notable in, for example, the zonal phases in February and July. This behaviour may result from a superposition of different tidal modes across the height range considered. However, there are also months where the ExUM and observed tidal phases are in good agreement (for example, the meridional phases in February or October).

Considering both of the large-amplitude tides, we see that there are times/heights of good agreement and times/heights where the agreement is less good. These biases may be a consequence of the simplified globally-uniform temperature nudging profile and the monthly fixed ozone background files used in this preliminary version of the ExUM. A move to a scheme with more realistic variation of temperature with latitude and season may therefore further improve tidal amplitudes in the ExUM. More fundamentally however, this globally-uniform nudging scheme needs to be replaced with molecular viscosity and diffusion (e.g., Griffin and Thuburn, 2018) as well as an improved chemistry scheme which will add the appropriate heating from exothermic reactions that is important throughout the thermosphere.

Yiğit and Medvedev (2017) and Miyoshi and Yiğit (2019) reported the migrating diurnal and semi-diurnal tidal amplitudes (respectively) and their interaction with GWs using the GW scheme of Yiğit et al. (2008). The diurnal tidal amplitudes in Yiğit and Medvedev (2017) in September of $10\text{–}30 \ \mathrm{ms}^{-1}$ in the zonal component and $10\text{–}50 \ \mathrm{ms}^{-1}$ in the meridional component agree reasonably with those observed in ExUM, where we see values of $10\text{–}30 \ \mathrm{ms}^{-1}$ in the zonal component and $15\text{–}45 \ \mathrm{ms}^{-1}$ in the meridional component. The semi-diurnal amplitudes in Miyoshi and Yiğit (2019) in June of $10\text{–}30 \ \mathrm{ms}^{-1}$ in the zonal component also agree with those observed in the ExUM, where we see values of $10\text{–}30 \ \mathrm{ms}^{-1}$ in the zonal component.

Dempsey et al. (2021) investigated diurnal and semi-diurnal tides over Rothera in the WACCM model and diurnal tides only in the eCMAM model. They also compared their results with meteor-radar observations, but their study considered only the year 2009 and so is not directly comparable to the results presented here for 2006. Nevertheless, the broad seasonal characteristics of the tides can be compared and some differences noted between the model results. Here we will again restrict our considerations to the large-amplitude semi-diurnal tide at Rothera, since the amplitude of the diurnal tide at this site is small in both models and observations.

The semi-diurnal tide predicted by both WACCM and the ExUM has monthly-mean amplitudes of c. $5\text{–}10 \ \mathrm{ms}^{-1}$ at heights of $80 \ \mathrm{km}$, which is generally comparable to the amplitudes revealed by the radar observations. However, above that height, although WACCM amplitudes increased with increasing height, they did so much less than is the case in the ExUM results presented here. In fact, the WACCM semi-diurnal tidal amplitudes exceeded $20 \ \mathrm{ms}^{-1}$ at a height of $100 \ \mathrm{km}$ in only two months in summer (November and December). In several months this matched well to the observations, but in other months was rather smaller than observed at the upper heights (March through to September). This contrasts with the much larger amplitudes evident in the ExUM at a height of $100 \ \mathrm{km}$, which we have shown are in the range $20\text{–}40 \ \mathrm{ms}^{-1}$ in all months except March and which in some months significantly exceeds the observed amplitudes (e.g., January and December). Inter-annual variability in tidal amplitude prevents a direct comparison, but it seems likely from this that semi-diurnal tidal amplitudes in the ExUM exceed those of WACCM at heights approaching $100 \ \mathrm{km}$ – at least in some months.

The WACCM results presented by Dempsey et al. (2021) also included estimates of monthly mean tidal phase as a function of height and indicated a good agreement in the phase gradients (i.e., vertical wavelength) between WACCM and observations in some summer and winter months (particularly, January, February, May–August and December), but less good agreement around the equinoxes. Similar behaviour is apparent in our ExUM results, although again, in some months the agreement is less good, e.g., meridional phases in May and July which suggest longer vertical wavelengths in the ExUM than observed.

Davis et al. (2013) investigated both diurnal and semi-diurnal tides over Ascension Island using data from the same meteor radar used in our study. They also compared their observations to results from both WACCM and eCMAM. However, they presented their results as averages for the entire interval 2002–2011 and so, again, the results are not directly comparable with those we report here. We will thus again restrict our comments to consideration of the broad seasonal characteristics of the large-amplitude diurnal tide at Ascension Island, since the amplitude of the semi-diurnal tide at this site is small in both models and observations.

In general, Davis et al. (2013) showed that eCMAM tended to overestimate the meridional amplitudes of the diurnal tide over Ascension Island, whereas WACCM tended to underestimate them. The differences were not so large in the case of the zonal component amplitudes. Both models predicted larger amplitudes at the upper heights considered. In contrast, the results we have presented here show that the monthly mean ExUM diurnal tidal amplitudes are not systematically larger or smaller than those observed, but from month to month can vary and be either larger or smaller.

Estimates of monthly mean tidal phase as a function of height and corresponding vertical wavelengths were also presented by Davis et al. (2013) for WACCM and eCMAM. Both models predicted tidal phases and vertical wavelengths with good agreement to the radar observations around the equinoxes, but with less good agreement in the summer and winter months (particularly eCMAM which predicted much shorter diurnal zonal vertical wavelengths than are observed in summer). The ExUM generally does well in predicting the diurnal tidal phases and phase gradients (i.e., vertical wavelength), but with some small differences in summer months.

## 5 Conclusions

We have presented the first study demonstrating the ability of the newly Extended Unified Model (ExUM) to capture the background winds and the atmospheric tides of the MLT. We have detailed the changes made to the model which allowed these investigations, including i) the addition of a non-LTE radiation scheme and ii) the relaxation to a climatological temperature profile above 90 km. We tested the predicted winds and tides in the ExUM by comparing them to the tides observed by SKiYMET meteor radars at characteristic Antarctic and equatorial latitudes where we expect the diurnal and semi-diurnal tides, respectively, to dominate. We used data from 2006 and for each month determined monthly-mean tidal amplitudes and phases.

Despite the simplified nature of this initial development of the ExUM, the model produces diurnal and semi-diurnal tides that display many characteristics of the observed tides. In particular, the monthly-mean amplitudes and vertical phase gradients are in reasonably good agreement with the observations in most months and at most heights. It is still true that in some months and at some heights the predicted tidal amplitudes can differ significantly from those observed. Given that the comparison of winds described above highlights limitations in the ExUM's gravity-wave parameterization, it may well be that this also impacts the model's tides and accounts for some of the differences.

1. The equatorial background MLT winds predicted by the ExUM capture some essential features well – the observed pattern of semi-annual variation is reproduced. However, there are several months where there are notable quantitative differences in the detail, e.g. Feb/Mar.

2. The polar background MLT winds predicted by the ExUM have some notable differences from those observed. Most striking are that i) the winds in the ExUM in austral summer are stronger than observed and ii) the observed eastward winds in austral winter are not reproduced in the model, which actually predicts westward winds.

3. We have proposed that these eastward winds in the real atmosphere are forced by the fluxes of non-primary GWs generated when large-amplitude orographic GWs break in the upper stratosphere or mesosphere, as suggested in the modelling study of Becker and Vadas (2018). These discrepancies between the model predictions and the observations highlight the limitations of gravity-wave parameterizations that only launch waves from near the surface.

4. The equatorial tidal amplitudes predicted by the ExUM are generally in good agreement with observations. Key qualitative features are reproduced, including large diurnal amplitudes and small semi-diurnal amplitudes; a general increase in amplitude with height; and the meridional tide component exceeding the zonal component.

5. The polar tidal amplitudes are generally good and also reproduce many of the qualitative features mentioned above. However, the ExUM noticeably overestimates the tidal amplitudes at the summer solstice. This is the height and time when the ExUM zonal winds are larger than those observed and we therefore propose the anomalous tidal amplitudes may be a consequence of these zonal winds.

6. The tidal phases of the larger tides have vertical phase gradients which are in very good agreement with observations.Key features are replicated including a general decrease in phase with height and the meridional phases leading their zonal counterparts. A difference in phase is commonly seen but is expected given the ground-level source of parameterized GWs.

It is necessary for high top models to reproduce these key features which are critical for deep coupling models as we strive towards more accurate models in the MLT. Further, we have suggested details for future work and parts of the model for future development. From this, we recommend two improvements to deal with the problems seen in the polar MLT, firstly the tuning of the spectral GW scheme to correct the wind direction in polar winter, and secondly reducing the magnitude of winds around 95 km in polar summer (which may in turn address the overly large tidal amplitudes observed in polar summer). These improvements pave the way for the development of a whole atmosphere UM in the near future.

In summary, we have demonstrated that even with relaxation to a relatively simplified temperature field and the use of monthly ozone background files, the ExUM can produce tides with many of the features observed, highlighting its usefulness for future tidal studies. Further, we have suggested that the ExUM's gravity-wave parameterization needs to be revised in light of what we infer to be the existence in the real atmosphere of significant fluxes of GWs not launched from the surface.

*Code availability.* The Unified Model code is provided courtesy of the UK Met Office and is subject to copyright.

*Data availability.* The Rothera and Ascension Island meteor radar data used in this study is from Mitchell, N., University of Bath and Moffat-Griffin, T., British Antarctic Survey: Rothera Skiymet Meteor Radar data (2005-present); Ascension Island Skiymet Meteor Radar data (2001-2011) and can be accessed at https://data.ceda.ac.uk/badc/meteor-radars. The model data is produced by the UK Met Office's Unified Model, copyright UK Met Office.

*Author contributions.* The experimental concept and design of methodology was performed by Griffith and Mitchell. The Rothera meteor radar is operated by British Antarctic Survey (with PI Moffat-Griffin). The radar data analysis was performed by Dempsey. The interpretation of results was performed by all authors. The final authorship of manuscript and preparation of figures was performed by Griffith.

*Competing interests.* No competing interests are present.

*Acknowledgements.* MJG and NJM are supported by a NERC GW4+ Doctoral Training Partnership studentship from the Natural Environment Research Council [NE/L002434/1] and are thankful for the collaborative support of the Met Office, UK. SMD is also supported by a NERC GW4+ Doctoral Training Partnership studentship from the Natural Environment Research Council [NE/L002434/1]. DRJ received funding for this work from the European Union Horizon 2020 research and innovation programme under grant agreement No. 776287. TMG

is supported by the Natural Environment Research Council [NE/R001391/1 and NE/R001235/1]. We are grateful to Chris Budd for helpful comments on an earlier version of this manuscript.

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
