# Peer review of "Winds and Tides of the Extended Unified Model in the Mesosphere and Lower Thermosphere Validated with Meteor Radar Observations"

_Annales Geophysicae, 2021_

## Referee Comment (RC2)

Review of
"Winds and Tides of the Extended Unified Model in the Mesosphere and Lower Thermosphere Validated with Meteor Radar Observations", Griffith et al.

The manuscript reports on a study of the atmospheric winds and tides with the Met Office's Extended Unified Model (ExUM), which extends from the lower atmosphere to the lower thermosphere. This model is based on a previous version of a Met Office model that has been recently extended to the mesosphere and lower thermosphere region. A comparison against meteor radar observation of winds and tides from 2006 between 80 and 100 km over two radar stations are performed. Specifically, the authors test the ability of the ExUM to model diurnal and semi-diurnal tides by comparing the model to observations of zonal and meridional winds made in the mesosphere and lower thermosphere. A standard least-square fitting method is used for tidal analysis.

The study shows that the ExUM model is capable of reproducing the overall structure of the atmospheric winds and tides, capturing many of the key characteristics seen in meteor radar observations, such as zonal and meridional wind maxima and minima, the increase in tidal amplitude with increasing height, and the decrease in tidal phase with increasing height. The model is still under development and further tests will certainly improve the GCM, which the authors correctly recognize.

I think, a development of an extended general circulation model is an important contribution to the field of atmospheric modeling and atmospheric coupling. Having more models connecting the lower atmosphere and the upper atmosphere will provide more opportunities for young researchers coming into the field of atmospheric vertical coupling. There is certainly room for further model development, however, model development is an iterative and long-term endeavor and the authors are in a good position to complete this. I would like to also mention that some important earlier contributions to the field of high-top GCMs and atmospheric wave coupling have not been included neither in the introduction nor in the discussion, so I encourage the authors to include in their discussion these contributions in the field of vertical coupling. For this, I have provided further details below. I suggest the manuscript to be published after the following comments are adequately addressed.

Comments:

1. In fact, high-top GCMs have been around for some time. Some earlier GCMs that extend from the lower atmosphere to the upper thermosphere have not been included in the high-top GCMs. Please add the following:

- The Coupled Middle Atmosphere Thermosphere-2 GCM extends from the lower atmosphere up to 300-500 km, depending on the solar activity. This GCM was first presented in the work by Yiğit et al. (2009) , which has utilized the nonlinear spectral GW parameterization of Yiğit et al. (2008) to study the propagation of a broad spectrum gravity waves from the lower atmosphere to the thermosphere. Note that this scheme is designed for the vertical evolution and dissipation of GWs. It is not a parameterization of GW sources. The most recent work with this

GCM is given in the work by Yiğit et al. (2021). The authors ought to include these studies in their introduction.

- The University of Leipzig Middle and Upper atmosphere model extends from the lower atmosphere up to 160 km. A recent study with this GCM on the interaction of GW and terdiurnal tides is given in the work by Lilienthal et al. (2021).

- The Kyushu GCM extending up to 450 km (S Miyahara et al., 1993; Miyoshi & Fujiwara, 2008; Miyoshi & Yiğit, 2019).

2. lines 116-118. The authors correctly acknowledge that modeling tides has been a challenging aspect of GCM studies. Without adequately modeling nonorographic GWs, tides cannot be properly simulated (S. Miyahara & Forbes, 1991). Please note here that the general circulation modeling study of Yiğit & Medvedev (2017) extensively discuss this aspect in the context of the coupling between the diurnal migrating tide and sub-grid-scale gravity waves. They show that GWs play an important role for the diurnal tide in the MLT region. They found that the GW effects on the thermal tide can be appropriately captured in a coarse-grid GCM provided that a GW parameterization (1) considers a broad spectrum of harmonics, (2) properly describes their propagation, and (3) correctly accounts for the physics of wave breaking/saturation.

3. Please discuss your results in the context of the recent publication of Miyoshi & Yiğit (2019) who have used the Kyushu GCM incorporating the nonlinear spectral GW scheme of Yiğit et al. (2008). They showed that the nonorographic subgrid-scale GWs attenuate the migrating semidiurnal solar-tide (SW2) amplitude in the lower thermosphere and modify the latitudinal structure of the SW2 above a 150 km height.

4. lines 159-161: What is the vertical extent of the damping coefficient? What kind of impact does it have on your results?

5. You may consider adding in the introduction the review paper by Yiğit et al. (2016), which discussed vertical coupling processes via internal waves, including, GWs and tides. This study has provided further motivation for the development of high-top models.

6. lines 206: What do you mean with year dependent forcing?

7. lines 206: Can you clarify whether you have used nudging above 90 km in this study or not. Why are the details postponed to another study? I think, they are relevant to this paper.

8. lines 534-538: I think, further details of the used GW parameterization should be provided in this study. A physics-based parameterization with an empirical GW source should include a broad spectrum of GWs and should be able to handle the MLT region without extensive tuning. Here are some helpful references that can guide your discussion: For example, the book chapter by Yiğit & Medvedev (2013) talks about the implementation methodology of an extended GW parameterization suitable for the middle and upper atmosphere. The work by Majdzadeh & Klaassen (2019) showed that the Warner-McIntyre scheme has in fact the ability to reproduce the observed spectral tail for the case of no background wind. However, it was also shown that the associated saturation threshold generally does not follow the observed spectral behavior at high vertical wavenumbers. Another insightful paper on GW dynamics in the middle atmosphere is the work by Geller et al. (2011). In a study of MLT tides, the GW aspects of the story should be carefully discussed.

9. As a follow-up to the discussion of GWs and possible missing physics: Have you included GW thermal effects? Previous studies, for example, by Hickey et al., (2011); Medvedev and Klaassen (2003); Yiğit & Medvedev (2009) have shown that GWs produce localized,

and occasionally very strong, heating and cooling, which certainly play an important role for the thermosphere and can also be important in the MLT region. Please discuss this aspect and consider including the cited contributions.

10. Please discuss your results in comparison with the GCM simulations of GW-tidal interactions performed by Yiğit and Medvedev (2017) for the GW-diurnal tides and Miyoshi and Yiğit (2019) for GW-semi-diurnal tide interactions. These studies have used the Yiğit et al. (2008) scheme, as noted above. Nonlinear interaction between GWs is an important process in the MLT region (Medvedev & Klaassen, 2000). Some of the biases in the simulations can be due to this missing process in ExUM. Please acknowledge these previous efforts adequately.

11. Yiğit et al. (2021) GCM showed that implementing a latitudinally variable GW spectrum in somewhat agreement with SABER observations can impact the middle atmosphere circulation significantly. This type of GW variability can have an important effect on the diurnal tides. This mechanism should be mentioned in the discussion.

**References:**

Geller, M. A., Zhou, T., Ruedy, R., Aleinov, I., Nazarenko, L., Tausnev, N. L., Sun, S., Kelley, M., & Cheng, Y. (2011). New Gravity Wave Treatments for GISS Climate Models. *Journal of Climate*, *24*(15), 3989–4002. https://doi.org/10.1175/2011JCLI4013.1

Hickey, M. P., Walterscheid, R. L., & Schubert, G. (2011). Gravity wave heating and cooling of the thermosphere: Sensible heat flux and viscous flux of kinetic energy. *Journal of Geophysical Research: Space Physics*, *116*(A12). https://doi.org/10.1029/2011JA016792

Lilienthal, F., Yiğit, E., Samtleben, N., & Jacobi, C. (2021). Variability of Gravity Wave Effects on the Zonal Mean Circulation and Migrating Terdiurnal Tide as Studied With the Middle and Upper Atmosphere Model (MUAM2019) Using a Nonlinear Gravity Wave Scheme. *Frontiers in Astronomy and Space Sciences*, *7*. https://doi.org/10.3389/fspas.2020.588956

Majdzadeh, M., & Klaassen, G. P. (2019). An analysis of the Hines and Warner–McIntyre–Scinocca non-orographic gravity wave drag parametrizations. *Quarterly Journal of the Royal Meteorological Society*, *145*(722), 2308–2334. https://doi.org/10.1002/qj.3559

Medvedev, A. S., & Klaassen, G. P. (2000). Parameterization of gravity wave momentum

    deposition based on nonlinear wave interactions: Basic formulation and sensitivity tests.

    *Journal of Atmospheric and Solar-Terrestrial Physics*, *62*(11), 1015–1033.

    https://doi.org/10.1016/S1364-6826(00)00067-5

Medvedev, A. S., & Klaassen, G. P. (2003). Thermal effects of saturating gravity waves in the

    atmosphere. *Journal of Geophysical Research*, *108*(D2), 4040.

    https://doi.org/10.1029/2002JD002504

Miyahara, S., & Forbes, J. M. (1991). Interactions between Gravity Waves and the Diurnal Tide

    in the Mesosphere and Lower Thermosphere. *Journal of the Meteorological Society of*

    *Japan. Ser. II*, *69*(5), 523–531. https://doi.org/10.2151/jmsj1965.69.5_523

Miyahara, S, Yoshida, Y., & Miyoshi, Y. (1993). Dynamic coupling between the lower and

    upper atmosphere by tides and gravity waves. *Journal of Atmospheric and Terrestrial*

    *Physics*, *55*(7), 1039–1053. https://doi.org/10.1016/0021-9169(93)90096-H

Miyoshi, Y., & Fujiwara, H. (2008). Gravity Waves in the Thermosphere Simulated by a General

    Circulation Model. *Journal of Geophysical Research*, *113*(D1), D01101.

    https://doi.org/10.1029/2007JD008874

Miyoshi, Y., & Yiğit, E. (2019). Impact of gravity wave drag on the thermospheric circulation:

    Implementation of a nonlinear gravity wave parameterization in a whole-atmosphere

    model. *Annales Geophysicae*, *37*(5), 955–969. https://doi.org/10.5194/angeo-37-955-

    2019

Yiğit, E., Aylward, A. D., & Medvedev, A. S. (2008). Parameterization of the effects of

    vertically propagating gravity waves for thermosphere general circulation models:

Sensitivity study. *Journal of Geophysical Research*, *113*(D19), D19106.
https://doi.org/10.1029/2008JD010135

Yiğit, E., Koucká Knížová, P., Georgieva, K., & Ward, W. (2016). A review of vertical coupling
in the Atmosphere–Ionosphere system: Effects of waves, sudden stratospheric warmings,
space weather, and of solar activity. *Journal of Atmospheric and Solar-Terrestrial
Physics*, *141*, 1–12. https://doi.org/10.1016/j.jastp.2016.02.011

Yiğit, E., & Medvedev, A. S. (2009). Heating and cooling of the thermosphere by internal
gravity waves. *Geophysical Research Letters*, *36*(14), L14807.
https://doi.org/10.1029/2009GL038507

Yiğit, E., & Medvedev, A. S. (2013). Extending the Parameterization of Gravity Waves into the
Thermosphere and Modeling Their Effects. In F.-J. Lübken (Ed.), *Climate and Weather
of the Sun-Earth System (CAWSES)* (pp. 467–480). Springer Netherlands.
https://doi.org/10.1007/978-94-007-4348-9_25

Yiğit, E., & Medvedev, A. S. (2017). Influence of parameterized small-scale gravity waves on
the migrating diurnal tide in Earth's thermosphere. *Journal of Geophysical Research:
Space Physics*, *122*(4), 4846–4864. https://doi.org/10.1002/2017JA024089

Yiğit, E., Medvedev, A. S., Aylward, A. D., Hartogh, P., & Harris, M. J. (2009). Modeling the
effects of gravity wave momentum deposition on the general circulation above the
turbopause. *Journal of Geophysical Research*, *114*(D7), D07101.
https://doi.org/10.1029/2008JD011132

Yiğit, E., Medvedev, A. S., & Ern, M. (2021). Effects of Latitude-Dependent Gravity Wave
Source Variations on the Middle and Upper Atmosphere. *Frontiers in Astronomy and
Space Sciences*, *7*. https://doi.org/10.3389/fspas.2020.614018

---

## Author Comment (AC1)

**Response to Reviewers: Comments on "Winds and Tides of the Extended Unified Model in the Mesosphere and Lower Thermosphere Validated with Meteor Radar Observations" by Griffith et al.**

Quotes from the reviewer are in bold, and responses are indented. We first wish to thank the reviewers for the insightful and useful comments provided on the first version of the manuscript. We have made the requested changes and believe the manuscript is strengthened as a result, particularly in regard to placing the work in the context of other GCM studies.

**Responses to Reviewer #1:**

**line (l) 17: Shouldn't it be "westward rather than eastward"?**

> Indeed it should. The text has been changed accordingly.

**l 43 or l 51 (or thereabouts). You might want to mention (or at least referenced) that migrating and non-migrating tides are generated in-situ in the thermosphere from the dissipation of GWs from deep convection mainly in the ITCZ (Vadas et al, 2014, JGR). This changes the TEC distributions, and could potentially modify the conditions for seeding equatorial plasma bubbles in the F region.**

> Agreed, thank you for this suggestion. From L41 onwards, we have added "Tides are also generated in-situ in the thermosphere from the dissipation of gravity waves cause by deep convection, primarily in the Intertropical Convergence Zone (ITCZ) (Vadas et al., 2014)...... [These tides] can also change the Total Electron Content (TEC) distributions, with the consequence of potentially modifying the conditions for seeding of equatorial plasma bubbles in the F-region"

**l 109: I believe you should add the HIAMCM as #8 to this high-top GCM list, since this GCM goes to z=450 km (Becker & Vadas, 2020, JGR). This GCM is unique in that it reproduces the TAD (traveling atmospheric disturbance) hotspot observed over the wintertime Southern Andes (e.g. Park et al, 2014, Trihn et al, 2018). These TADs are composed of tertiary GWs from MW breaking at lower altitudes. The HIAMCM only models the neutrals from dynamics from below, and is a high-res GW-resolving model down to horizontal wavelengths of 165 km. The tides in the thermosphere agree pretty well with MSIS and HWM14 (Becker & Vadas, 2020), and this model includes ion drag, molecular viscosity, non-hydrostatic corrections. The model has explicit moisture cycle and radiative transfer. However the methods used are idealized compared to comprehensive GCMs. Additionally the HIAMCM has no chemistry. For these reasons, the word "Mechanistic" is included in the name of the model (HIAMCM= (HI Altitude Mechanistic general Circulation Model). Note that when the HIAMCM had a top of z~100 km, the tides (amplitudes and phases) were compared to data (Becker, 2017, JAS). At that time, the model was called KMCM, was lower resolution, and only had a GW parameterization scheme. However, the moisture and radiative transfer methods were essentially the same as in the HIAMCM.**

> Thank you for this suggestion and it is a valid point. HIAMCM has been added as #8 to the list of high-top models in a similar style to that used for the other models.

**Somewhere in the introduction the authors should mention an important recently-published paper by Stober et al (2021, AG). This paper examine the mean winds and diurnal and semidiurnal tidal amplitude and phases (and momentum fluxes) obtained from meteor radar data at six Southern Hemisphere locations (midlatitude to polar). They found that the results agreed reasonably well with Becker and**

**Vadas (2018, JGR), thereby pointing to secondary GWs and multistep vertical coupling as a mechanism by which GWs transfer energy and momentum to higher altitudes during the wintertime.**

Agreed, thank you for this suggestion. We have added text at L165 which highlights this study and its key results, noting in particular that it supports the work performed in Becker and Vadas (2018). A sentence has been added to the discussion referencing this useful work which further supports the proposal of secondary GWs being an important factor in the MLT.

**x-axis and y-axis number and labels for Figs 4-7, 16-17:**

**These numbers and labels are virtually unreadable (too small).  Please make them 2-3 times larger.**

Agreed. The numbers and labels have been made larger.

**l 261: sentence is confusing.  remove "than those observed"?**

Text changed appropriately.

**l 262: "and -40 m/s at heights near 80 km in january".  I don't see this in fig 6a?**

The text mistakenly referred to model winds instead of observed winds where this is the case. Text changed appropriately.

**l 320: "Sep., Oct."**

The sentence has been adjusted so that it does not start with "Sep" which was confusing.

**l 337: "at for"**

Text changed appropriately.

**Responses to Reviewer #2:**

**Comments:**

**1) In fact, high-top GCMs have been around for some time. Some earlier GCMs that extend from the lower atmosphere to the upper thermosphere have not been included in the list of high-top GCMs. Please add the following:**

> **a) The Coupled Middle Atmosphere Thermosphere-2 GCM extends from the lower atmosphere up to 300-500 km, depending on the solar activity. This GCM was first presented in the work by Yiğit et al. (2009), which has utilized a nonlinear spectral GW parameterization of Yiğit et al. (2008) to study the propagation of a broad spectrum gravity waves from the lower atmosphere to the thermosphere. Note that this scheme is designed for the vertical evolution and dissipation of GWs. It is not a parameterization of GW sources. The most recent work with this GCM is given in the work by Yiğit et al. (2021). The authors ought to include these studies in their introduction.**

> **b) The University of Leipzig Middle and Upper atmosphere model extends from the lower atmosphere up to 160 km. A recent study with this GCM on the interaction of GW and terdiurnal tides is given in the work by Lilienthal et. al (2021).**

> **c)The Kyushu GCM extending up to 450 km (Miyahara et al, 1993; Miyoshi and Fujiwara, H., 2008; Miyoshi and Yiğit, 2019).**

>> Thank you for bringing these to our attention. They have been added to the list of high-top GCMs. Also, a paragraph has been added on L146 discussing parameterization of non-orographic GWs (see below). These changes provide useful further context.

**2) Lines 116-118. The authors correctly acknowledge that modeling tides has been a challenging aspect of GCM studies. Without adequately modeling nonorographic GWs, tides cannot be properly simulated (Miyahara and Forbes, 1991). Please note here that the general circulation modeling study of Yiğit and Medvedev (2017) extensively discuss this aspect in the context of the coupling between the diurnal migrating tide and sub-grid-scale gravity waves. They show that GWs play an important role for the diurnal tide in the MLT region. They found that the GW effects on the thermal tide can be appropriately captured in a coarse-grid GCM provided that a GW parameterization (1) considers a broad spectrum of harmonics, (2) properly describes their propagation, and (3) correctly accounts for the physics of wave breaking/saturation.**

> This is an important aspect which we had previously overlooked. A paragraph discussing parameterization of non-orographic GWs has been added on L146.

**3) Please discuss your results in the context of the recent publication of Miyoshi and Yiğit (2019) who have used the Kyushu GCM incorporating the nonlinear spectral GW scheme of Yiğit et al. (2008). They showed that the nonorographic subgrid-scale GWs attenuate the migrating semidiurnal solar-tide (SW2) amplitude in the lower thermosphere and modify the latitudinal structure of the SW2 above a 150 km height.**

> We have looked at the model fields and do indeed notice that the USSP acts to attenuate semi-diurnal tidal amplitudes under solstice conditions. A paragraph has been added to the discussion on L612 to address this subject.

**4) Lines 159-161: What is the vertical extent of the damping coefficient? What kind of impact does it have on your results?**

The vertical extent of the damping coefficient is discussed in more detail in Griffith et al. (2020) – we have added a reference to this and an explanation to the text. In summary, the vertical damping coefficient acts to damp the velocity of the vertical wind in the model, which begins weakly half-way through the model vertical domain and strengthens to a prescribed value at the top of the model. The damping coefficient extends to the ground around the poles for model stability around the node of the latitude-longitude mesh. The primary impact of the damping coefficient is to reduce the magnitude of instantaneous vertical velocities approaching the upper boundary which can lead to model instabilities – this will ideally be replaced with molecular diffusion in future versions of the model. The damping coefficient is therefore chosen so that the model can run in a stable manner (also detailed in Griffith et al. 2020). A particularly large value of the vertical damping coefficient can result in consequent damping of horizontal wind velocities in the upper layers of the model which is why we chose to avoid using the 120 and 135 km implementations from the previous work. We have included a few sentences to explain this in the paper around L200.

**5) You may consider adding in the introduction the review paper by Yiğit et al. (2016), which discussed vertical coupling processes via internal waves, including, GWs and tides. This study has provided further motivation for the development of high-top models.**

Thank you for bringing this review to our attention. We agree this is a useful review and have included this work in the introduction around L73.

**6) Lines 206: What do you mean with year dependent forcing?**

By this we mean forcing of atmospheric fields to year dependent atmospheric data (such as reanalyses like ERA-Interim from the ECMWF), so that the model has background forcing in the lower atmosphere which is dependent on how the atmosphere behaved in a specific year. We do not use this type of forcing. This is the difference between a "free-running" and "specified-dynamics" model run, such as, for example, Specified Dynamics (SD) WACCM-X used in Liu et al., (2018). We have changed the text around L250 and included the below citation to clarify this point.

Liu, J., Liu, H., Wang, W., Burns, A. G., Wu, Q., Gan, Q., ... & Schreiner, W. S. (2018). First results from the ionospheric extension of WACCM-X during the deep solar minimum year of 2008. Journal of Geophysical Research: Space Physics, 123(2), 1534-1553.

**7) Lines 206: Can you clarify whether you have used nudging above 90 km in this study or not. Why are the details postponed to another study? I think, they are relevant to this paper.**

Apologies, the phrasing here is ambiguous. We have used nudging above 90 km in this study. It accounts for the lack of appropriate high atmosphere chemistry and consequent heating via exothermic reactions not present in the model (mentioned in L170 of the first draft). We shall remove the previous reference to the nudging here – this sentence was to reiterate that the nudging is simplified in nature, but this is mentioned several times previously. The nudging region is shown in Figure 2 and discussed in detail in Griffith et al. (2020) where it was implemented. Reference to this is made in the text, beginning around L196 with the mention of the relaxation scheme and continuing to L215 where we specify that the nudging is still used above 90 km. We have made additions for clarification in the text and hope it resolves this confusion.

**8) Lines 534-538: I think, further details of the used GW parameterization should be provided in this study. A physics-based parameterization with an empirical GW source should include a broad spectrum of GWs and should be able to handle the MLT region without extensive tuning. Here are some helpful references that can guide your discussion: For example, the book chapter by Yiğit and Medvedev (2013) talks about the implementation methodology of an extended GW parameterization suitable for the middle and upper atmosphere. The work by Majdzadeh and Klaassen (2019) showed that the Warner-McIntyre scheme has in fact the ability to reproduce the observed spectral tail for the case of no background wind. However, it was also shown that the associated saturation threshold generally does not follow the observed spectral behavior at high vertical wavenumbers. Another insightful paper on GW dynamics in the middle atmosphere is the work by Geller et al. (2011). In a study of MLT tides, the GW aspects of the story should be carefully discussed.**

> We agree that this is an important point. Two paragraphs have been added giving more detail on the GW parameterization used, starting at L583 (and relating also to point (9)). While the USSP appears to be doing a reasonable job in the MLT, additional mechanisms will no doubt be necessary in this region, such as in-situ GW generation and latitudinally varying the spectrum as you mention in point (11) below. We try to reflect this in our comments here. A more complete study such as that performed by Yiğit et al. (2008) is needed to evaluate the capabilities of the USSP in the MLT. A detailed discussion of the current state and future developments of the USSP is beyond the scope of this study.

**9) As a follow-up to the discussion of GWs and possible missing physics: Have you included GW thermal effects? Previous studies, for example, by Medvedev and Klaassen (2003), Yiğit and Medvedev (2009), and Hickey et al. (2011) have shown that GWs produce localized, and occasionally very strong, heating and cooling, which certainly play an important role for the thermosphere and can also be important in the MLT region. Please discuss this aspect and consider including the cited contributions.**

> Some GW thermal affects are included, but the model does not include ionospheric heating effects such as ion drag, and we now refer to this fact in the text, starting at L591. Namely, the GW scheme includes the frictional heating due to gravity wave dissipation, and consequent loss of kinetic energy:
>
> $$\frac{\partial T}{\partial t} = -\frac{1}{c_p}\left(u\frac{\partial u}{\partial t} + v\frac{\partial v}{\partial t}\right)$$
>
> where $T$ is the temperature, $c_p$ is the specific heat capacity at constant pressure and the $\partial/\partial t$ terms are the total tendencies due to both the orographic and non-orographic GW drag schemes. More details can be found in Sect. 2.7 and 3.5 of Walters et al., 2017 and an additional reference to this is added in this part of the text. As well as this, we now include the cited contributions to show the importance of GW heating (around L594).

**10) Please discuss your results in comparison with the GCM simulations of GW-tidal interactions performed by Yiğit and Medvedev (2017) for the GW-diurnal tides and Miyoshi and Yiğit (2019) for GW-semi-diurnal tide interactions. These studies have used the Yiğit et al. (2008) scheme, as noted above. Nonlinear interaction between GWs is an important process in the MLT region (Medvedev and Klaassen, 2000). Some of the biases in the simulations can be due to this missing process in ExUM. Please acknowledge these previous efforts adequately.**

We now reference these useful works. We observe similar amplitudes to those reported in the 80 – 100 km region. A paragraph providing a quantitative comparison with the diurnal and semi-diurnal amplitudes observed therein with those observed in our results has been added at L650.

**11) Yiğit et al. (2021) GCM showed that implementing a latitudinally variable GW spectrum in somewhat agreement with SABER observations can impact the middle atmosphere circulation significantly. This type of GW variability can have an important effect on the diurnal tides. This mechanism should be mentioned in the discussion.**

Agreed. We have added a paragraph commenting on the impact of a latitudinally varying GW spectrum and its priority as a future development in the USSP on L595, with reference to the cited contributions.